# Leave-One-Out Stable Conformal Prediction

**Kiljae Lee, Yuan Zhang**
The Ohio State University
`lee.10428@osu.edu, yzhanghf@stat.osu.edu`

## Abstract

Conformal prediction (CP) is an important tool for distribution-free predictive uncertainty quantification. Yet, a major challenge is to balance computational efficiency and prediction accuracy, particularly for multiple predictions. We propose **Leave-One-Out Stab**le **C**onformal **P**rediction (`LOO-StabCP`), a novel method to speed up full conformal using algorithmic stability without sample splitting. By leveraging *leave-one-out* stability, our method is much faster in handling a large number of prediction requests compared to existing method `RO-StabCP` based on *replace-one* stability. We derived stability bounds for several popular machine learning tools: regularized loss minimization (RLM) and stochastic gradient descent (SGD), as well as kernel method, neural networks and bagging. Our method is theoretically justified and demonstrates superior numerical performance on synthetic and real-world data. We applied our method to a screening problem, where its effective exploitation of training data led to improved test power compared to state-of-the-art method based on split conformal.

## 1 Introduction

*Conformal prediction* (CP) offers a powerful framework for distribution-free prediction. It is useful for a variety of machine learning tasks and applications, including computer vision (Angelopoulos et al., 2020), medicine (Vazquez & Facelli, 2022; Lu et al., 2022), and finance (Wisniewski et al., 2020), where reliable uncertainty quantification for complex and potentially mis-specified models is in much need. Initially introduced by Vovk et al. (2005), conformal prediction has gained renewed interest. Numerous studies significantly enriched the conformal prediction toolbox and deepened theoretical understandings (Lei et al., 2018; Gibbs & Candes, 2021; Barber et al., 2023).

Given data $\mathcal{D} = \{(X_i, Y_i)\}_{i=1}^n$, where $(X_i, Y_i) \in (\mathcal{X}, \mathcal{Y}) \overset{\text{i.i.d.}}{\sim} P_{X,Y}$, the goal is to predict the unobserved responses $\{Y_{n+j}\}_{j=1}^m$ on the *test data* $\mathcal{D}_{\text{test}} = \{(X_{n+j}, Y_{n+j}{=}?)\}_{j=1}^m \overset{\text{i.i.d.}}{\sim} P_{X,Y}$. Conformal prediction constructs prediction intervals $\mathcal{C}_\alpha(X_{n+j})$ at any given level $\alpha \in (0,1)$, such that

$$\mathbb{P}(Y_{n+j} \in \mathcal{C}_\alpha(X_{n+j})) \geq 1 - \alpha, \quad \text{for all } j = 1, \ldots, m. \tag{1}$$

A highlighted feature of conformal prediction is *distribution-free*: even when a wrong model is used for prediction, the coverage validity (1) still holds (but the prediction interval will become wider).

A primary challenge in conformal prediction lies in balancing computation cost with prediction accuracy. Among the variants of conformal prediction, *full conformal* is the most accurate (i.e., shortest predictive intervals) but also the slowest; *split conformal* greatly speeds up by a data-splitting scheme, but decreases accuracy and introduces variability that heavily depends on the particular split (Angelopoulos & Bates, 2021; Vovk, 2015; Barber et al., 2021). Derandomization (Solari & Djordjilović, 2022; Gasparin & Ramdas, 2024) addresses the latter issue but increases computational cost and may make the method more conservative (Ren & Barber, 2024).

Algorithmic stability is an important concept in machine learning theory (Bousquet & Elisseeff, 2002). It measures the sensitivity of a learning algorithm to small changes in the input data. Numerous studies have focused on techniques that induce algorithmic stability, such as regularized loss minimization (Shalev-Shwartz et al., 2010; Shalev-Shwartz & Ben-David, 2014) and stochastic gradient descent (Hardt et al., 2016; Bassily et al., 2020). Recent research has applied the concept of algorithmic stability to the field of conformal prediction (Ndiaye, 2022; Liang & Barber, 2023).

Ndiaye (2022) proposed *replace-one stable conformal prediction* (`RO-StabCP`) that effectively accelerates full conformal by leveraging algorithmic stability. While it accelerates full conformal without introducing data splitting, thus preserving prediction accuracy, the prediction model needs to be refit for each $X_{n+j}$, lowering its computational efficiency for multiple predictions.

In this paper, we introduce **L**eave-**O**ne-**O**ut **Stab**le **C**onformal **P**rediction (`LOO-StabCP`), which represents is a distinct form of algorithmic stability type than that in `RO-StabCP`. The key innovation is that our stability correction no longer depends on the predictor at the test point $X_{n+j}$. As a result, our method only needs *one* model fitting using the training data $\mathcal{D}$. This enables our method to effectively handle a large number of prediction requests. Theoretical and numerical studies demonstrate that our method achieves competitive prediction accuracy compared to existing method, while preserving the finite-sample coverage validity guarantee.

## 2 PRIOR WORKS ON CONFORMAL PREDICTION (CP)

To set up notation and introduce our method, we begin with a brief review of classical CP methods.

**Full conformal.** We begin by considering the prediction of a single $Y_{n+j}$. Fix $j \in [m] = \{1, \ldots, m\}$ and let $y$ denote a guessed value of the unobserved $Y_{n+j}$. We call $\mathcal{D}_j^y = \mathcal{D} \cup \{(X_{n+j}, y)\}$ the augmented data and train a learning algorithm $f$ (such as linear regression) on $\mathcal{D}_j^y$. To emphasize that the outcome of the training depends on both $X_{n+j}$ and $y$, we denote the fitted model by $\widehat{f}_j^y$. Here we require that the training algorithm is permutation-invariant, meaning that $\widehat{f}_j^y$ remains unchanged if any two data points $(X_i, Y_i)$ and $(X_{i'}, Y_{i'})$ are swapped for $i, i' \in [n] \cup \{n + j\}$. Next, for each $i = 1, \ldots, n, n + j$, we define a *non-conformity score* $S_{i,j}^y = S(Y_i, \widehat{f}_j^y(X_i)) = |Y_i - \widehat{f}_j^y(X_i)|$ to measure $\widehat{f}_i^y$'s goodness of prediction on the $i$th data point. Notice that $S_{i,j}^y$ also depends on $X_{n+j}$ through $\widehat{f}_j^y(X_i)$, thus its subscript $j$. For simplicity, we set $j = 1$ and write $S_{i,j}^y$ as $S_i^y$ only for this part. Now, plugging in $y = Y_{n+1}$, by symmetry, all non-conformity scores $\{S_i^{Y_{n+1}}\}_{i=1}^n \cup \{S_{n+1}^{Y_{n+1}}\}$ are exchangeable, and then the rank of $S_{n+1}^{Y_{n+1}}$ (in ascending order) is uniformly distributed over $\{1, \ldots, n + 1\}$, implying that

$$\mathbb{P}(S_{n+1}^{Y_{n+1}} \leq \mathcal{Q}_{1-\alpha}(\{S_i^{Y_{n+1}}\}_{i=1}^n \cup \{\infty\})) \geq 1 - \alpha,$$

[1] where $\mathcal{Q}_p(\mathcal{S}) := \inf\{x | F_{\mathcal{S}}(x) \geq p\}$ denotes the lower-$p$ sample quantile of data $\mathcal{S}$. This implies coverage validity $\mathbb{P}(Y_{n+1} \in \mathcal{C}_\alpha^{\text{full}}(X_{n+1})) \geq 1 - \alpha$ of the prediction set $\mathcal{C}_\alpha^{\text{full}}(X_{n+1})$ defined as

$$\mathcal{C}_\alpha^{\text{full}}(X_{n+1}) = \left\{y \in \mathcal{Y} : S_{n+1}^y \leq \mathcal{Q}_{1-\alpha}(\{S_i^y\}_{i=1}^n \cup \{\infty\})\right\}. \quad (2)$$

This leads to the *full conformal* (`FullCP`) method: compute $\mathcal{C}_\alpha^{\text{full}}(X_{n+1})$ by a grid search over $\mathcal{Y}$.

**Split conformal.** The grid search required by full conformal is expensive. The key to acceleration is to decouple the prediction function $\widehat{f}$, thus most non-conformity scores $\{S_i^y\}_{i=1}^n$, from both $j$ and $y$: if $S_{n+1}^y$ is the only term that depends on $y$, then the prediction set can be analytically solved from (2). *Split conformal* (`SplitCP`) (Papadopoulos et al., 2002; Vovk, 2015) randomly splits $\mathcal{D}$ into the training data $\mathcal{D}_{\text{train}}$ and the calibration data $\mathcal{D}_{\text{calib}}$, train $\widehat{f}$ only on $\mathcal{D}_{\text{train}}$, and compute and rank non-conformity scores only on $\mathcal{D}_{\text{calib}} \cup \{(X_{n+j}, y)\}$. While split conformal effectively speeds up computation, the flip side is the reduced amount of data used for both training and calibration, leading to wider predictive intervals and less stable output.

**Replace-one stable conformal.** Ndiaye (2022) accelerated `FullCP` by leveraging algorithmic stability. From now on, we will switch back to the full notation for $S$ and no longer abbreviate $S_{i,j}^y$ as $S_i^y$. To decouple the non-conformity scores from $y$, Ndiaye (2022) evaluate these scores using $\widetilde{y}$, an arbitrary guess of $y$. Therefore, we call his method *replace-one stable conformal prediction* (`RO-StabCP`). To bound the impact of guessing $y$, he introduced the replace-one (RO) stability.

**Definition 1** (Replace-One Algorithmic Stability). *A prediction method $\widehat{f}$ is **replace-one stable**, if for all $j \in [m]$ and $i \in [n] \cup \{n + j\}$, there exists $\tau_{i,j}^{\text{RO}} < \infty$, such that*

$$\sup_{y, \widetilde{y}, z \in \mathcal{Y}} |S(z, \widehat{f}_j^y(X_i)) - S(z, \widehat{f}_j^{\widetilde{y}}(X_i))| \leq \tau_{i,j}^{\text{RO}},$$

---

[1]Here we replaced "$S_{n+1}^\infty$" in $\{S_i^{Y_{n+1}}\}_{i=1}^n \cup \{S_{n+1}^\infty\}$ by $\infty$. This does not change the quantile.

where $\widehat{f}_j^{\mathfrak{y}}$ is trained on $\mathcal{D} \cup \{(X_{n+j}, \mathfrak{y})\}$, for $\mathfrak{y} = y$ or $\widetilde{y}$.

Recall that $S_{i,j}^{\widetilde{y}} = |Y_i - \widehat{f}_j^{\widetilde{y}}(X_i)|$ denote the non-conformity score computed using $\widetilde{y}$. Then it suffices to build a predictive interval that contains $\mathcal{C}_{j,\alpha}^{\mathrm{full}}(X_{n+j})$ in (2). By Definition 1, the following inequality $|y - \widehat{f}_j^{\widetilde{y}}(X_{n+j})| - \tau_{n+j,j}^{\mathrm{RO}} \leq S_{n+j,j}^y \leq \mathcal{Q}_{1-\alpha}(\{S_{i,j}^y\}_{i=1}^n \cup \{\infty\}) \leq \mathcal{Q}_{1-\alpha}(\{S_{i,j}^{\widetilde{y}} + \tau_{i,j}^{\mathrm{RO}}\}_{i=1}^n \cup \{\infty\})$ holds true for any $y \in \mathcal{C}_{j,\alpha}^{\mathrm{full}}(X_{n+j})$. Consequently, the RO stable prediction set

$$\mathcal{C}_{j,\alpha}^{\mathrm{RO}}(X_{n+j}) = \left\{ y \in \mathcal{Y} : |y - \widehat{f}_j^{\widetilde{y}}(X_{n+j})| - \tau_{n+j,j}^{\mathrm{RO}} \leq \mathcal{Q}_{1-\alpha}(\{S_{i,j}^{\widetilde{y}} + \tau_{i,j}^{\mathrm{RO}}\}_{i=1}^n \cup \{\infty\}) \right\} \quad (3)$$

contains $\mathcal{C}_{j,\alpha}^{\mathrm{full}}(X_{n+j})$ as a subset, thus also has valid coverage. The numerical studies in Ndiaye (2022) demonstrated that `RO-StabCP` computes as fast as `SplitCP` while stably producing more narrower predictive intervals (i.e., higher prediction accuracy).

# 3 LOO-STABCP: LEAVE-ONE-OUT STABLE CONFORMAL PREDICTION

## 3.1 LEAVE-ONE-OUT (LOO) STABILITY AND GENERAL FRAMEWORK

When predicting one $Y_{n+j}$, `RO-StabCP` has accelerated full conformal to the speed comparable to split conformal. However, its non-conformity scores $S_{i,j}^{\widetilde{y}}$'s still depend on $X_{n+j}$. Consequently, in order to predict $\{Y_{n+j}\}_{j=1}^m$, `RO-StabCP` would refit the model $m$ times, once for each $X_{n+j}$.

This naturally motivates our approach: can we let all predictions be based off a *common* model $\widehat{f}$, which only depends on $\mathcal{D} = \{(X_i, Y_i)\}_{i=1}^n$, but not any of $\{X_{n+j}\}_{j=1}^m$? Interestingly, the idea might appear similar to a beginner's mistake when learning `FullCP`, overlooking that the model fitting in `FullCP` should also include $(X_{n+j}, y)$, not just $\mathcal{D}$. Clearly, to ensure a valid method, we must correct for errors inflicted by using $\widehat{f}$ in lieu of $\widehat{f}_j^y$. Since these two model fits (ours vs `FullCP`) are computed on similar sets of data, with the only difference of whether to consider $(X_{n+j}, y)$, our strategy is to study the *leave-one-out (LOO) stability* of the prediction method.

**Definition 2** (Leave-One-Out Algorithmic Stability). *A prediction method is **leave-one-out stable**, if for all $j \in [m]$ and $i \in [n] \cup \{n+j\}$, there exists $\tau_{i,j}^{\mathrm{LOO}} < \infty$, such that*

$$\sup_{y,z \in \mathcal{Y}} |S(z, \widehat{f}_j^y(X_i)) - S(z, \widehat{f}(X_i))| \leq \tau_{i,j}^{\mathrm{LOO}}.$$

The $\tau_{i,j}^{\mathrm{LOO}}$'s appearing in Definition 2 are called *LOO stability bounds*. Their values can often be determined by analysis of the specific learning algorithm $f$. For each $j$, we used a different set of LOO stability bounds $\{\tau_{i,j}^{\mathrm{LOO}}\}_{i \in [n] \cup \{n+j\}}$. This approach is adopted to achieve sharper bounds compared to using a uniformly bound for all $j$. We clarify that the concept of algorithmic stability is well-defined for parametric or non-parametric $f$'s alike. For an $f$ parameterized by some $\theta$, the stability bound does not focus on the whereabout of the optimal $\theta$, but on how much impact leaving out $(n+j)$th data point will have on the trained $f$, possibly via quantifying its impact on the estimated $\theta$. We will elaborate using concrete examples in Section 3.2. For now, we assume that $\tau_{i,j}^{\mathrm{LOO}}$'s are known and present the general framework of our method, called *leave-one-out stable conformal prediction* (`LOO-StabCP`), as Algorithm 1.

The implementation requires computation of $\mathcal{O}(mn)$ many $\tau_{i,j}^{\mathrm{LOO}}$ values. However, these computations are relatively inexpensive and do not significantly impact the overall time cost. In many examples (such as SGD, see Section 3.2.2), the main computational cost comes from model fitting, especially for complex models. We empirically confirm this in Section 4 through various numerical experiments.

Table 1 presents a comparison of the computational complexity for conformal prediction methods. The concept of leave-one-out perturbations in conformal prediction has been studied in Liang & Barber (2023), but their angle is very different from ours. They focused on studying the LOO as a part of `Jackknife+` (Barber et al., 2021), which fits $n$ models, one for each $\mathcal{D} \setminus \{(X_i, Y_i)\}$. Then all these $n$ models are used simultaneously for each prediction. In contrast, we use LOO technique in a very different way, developing a fast algorithm that fit *only one* model to $\mathcal{D}$ (without deletion). The "one" in our leave-one-out refers to "$(X_{n+j}, y)$" in $\mathcal{D} \cup \{(X_{n+j}, y)\}$, for each $j$.

---

**Algorithm 1:** (`LOO-StabCP`) Leave-One-Out Stable Conformal Prediction Set

---

**Input** : Training set $\mathcal{D} = \{(X_i, Y_i)\}_{i=1}^n$, test points $\{X_{n+j}\}_{j=1}^m$, desired coverage $1 - \alpha$.
**Output:** Prediction interval $\mathcal{C}_{j,\alpha}^{\mathrm{LOO}}(X_{n+j})$ for each $j \in [m]$

1. Fit the prediction function $\widehat{f}$ on $\mathcal{D}$;
2. Compute (uncorrected) non-conformity scores on $\mathcal{D}$: $S_i = |Y_i - \widehat{f}(X_i)|$ for $i \in [n]$;
**for** $j \in [m]$ **do**
    3. Compute stability bounds $\tau_{i,j}^{\mathrm{LOO}}$ for $i \in [n] \cup \{n + j\}$;
    4. Compute prediction interval:
$$\mathcal{C}_{j,\alpha}^{\mathrm{LOO}}(X_{n+j}) = \left[\widehat{f}(X_{n+j}) \pm \left\{\mathcal{Q}_{1-\alpha}\big(\{S_i + \tau_{i,j}^{\mathrm{LOO}}\}_{i=1}^n \cup \{\infty\}\big) + \tau_{n+j,j}^{\mathrm{LOO}}\right\}\right];$$
**end**

---

| | FullCP | SplitCP | RO-StabCP | **LOO-StabCP** |
|---|---|---|---|---|
| # of model fits | $|\mathcal{Y}| \cdot m$ | 1 | $m$ | **1** |
| # of prediction evaluations | $(n+1) \cdot |\mathcal{Y}| \cdot m$ | $n + m$ | $(n+1) \cdot m$ | $\boldsymbol{n + m}$ |
| # of stability bounds | Not applicable | Not applicable | $(n+1) \cdot m$ | $\boldsymbol{(n+1) \cdot m}$ |

Table 1: Computational complexities of our method and benchmarks, where $|\mathcal{Y}|$ is the size of the search grid used in FullCP. We emphasize that in many examples, such as SGD (Section 3.2.2), one model fitting is much more costly than one prediction or computation of one stability bound.

Next, we provide the theoretical guarantee of our algorithm's coverage validity.

**Theorem 1.** *If the prediction method is leave-one-out stable as in Definition 2. Then for each $j \in [m]$, the prediction set $\mathcal{C}_{j,\alpha}^{\mathrm{LOO}}(X_{n+j})$ constructed by Algorithm 1 satisfies*

$$\mathbb{P}(Y_{n+j} \in \mathcal{C}_{j,\alpha}^{\mathrm{LOO}}(X_{n+j})) \geq 1 - \alpha.$$

### 3.2 LOO Stable Algorithms

So far, we have been treating the stability bounds $\tau_{i,j}^{\mathrm{LOO}}$ as given without showing how to obtain them. In this section, we derive these bounds for two important machine learning tools: *Regularized Loss Minimization* (RLM) and *Stochastic Gradient Descent* (SGD). Many machine learning tasks aim to minimize a loss function $\ell(y, f_\theta(x))$ over training data. *Empirical Risk Minimization* (ERM) is a common approach, which seeks to minimize $\frac{1}{n} \sum_{i=1}^n \ell(Y_i, f_\theta(X_i))$ with respect to $\theta$. However, the objective function is often highly nonconvex, making the optimization challenging. RLM alleviates nonconvexity by adding an *explicit* penalty (e.g., ridge and LASSO) to the objective function (Hoerl & Kennard, 1970; Tibshirani, 1996). Alternatively, SGD *implicitly* regularizes the optimization procedure (Robbins & Monro, 1951) by iteratively updating model parameters using one data point at a time. Its computational efficiency makes it a preferred method in deep learning (LeCun et al., 2015; He et al., 2016).

#### 3.2.1 Example 1: Regularized Loss Minimization (RLM)

To derive the LOO stability bound, we compare two versions of RLM, only differing by their training data. The first is trained on $\mathcal{D}$, producing $\widehat{\theta} = \arg\min_{\theta \in \Theta}[\frac{1}{n}\sum_{i=1}^n \ell(Y_i, f_\theta(X_i)) + \Omega(\theta)]$, where $\Theta$ is the parameter space and $\Omega(\theta)$ is the explicit penalty term; while the second is trained on the augmented data $\mathcal{D}_j^y$ (recall $\mathcal{D}_j^y = \mathcal{D} \cup \{(X_{n+j}, y)\}$), producing $\widehat{\theta}_j^y = \arg\min_{\theta \in \Theta}[\frac{1}{n+1}\{\sum_{i=1}^n \ell(Y_i, f_\theta(X_i)) + \ell(y, f_\theta(X_{n+j}))\} + \Omega(\theta)]$. The LOO stability for RLM is described by Definition 2, with $\widehat{f}(\cdot) = f_{\widehat{\theta}}(\cdot)$ and $\widehat{f}_j^y(\cdot) = f_{\widehat{\theta}_j^y}(\cdot)$. To state our main result, we need some concepts from optimization.

**Definition 3** ($\rho$-Lipschitz). *A continuous function $g : \mathbb{R}^p \to \mathbb{R}^q$ is $\rho$-**Lipschitz**, if*

$$\|g(x) - g(y)\| \leq \rho\|x - y\|, \quad \text{for any } x, y \in \mathbb{R}^p.$$

**Definition 4** (Strong Convexity). *A function $g : \mathbb{R}^p \to \mathbb{R}$ is $\lambda$-**strongly convex**, if*

$$g(tx + (1-t)y) \leq tg(x) + (1-t)g(y) - \frac{\lambda}{2}\|x - y\|^2, \quad \text{for any } x, y \in \mathbb{R}^p \text{ and } t \in (0,1).$$

*In addition, a function $g$ is **convex** if it is 0-strongly convex.*

Now we are ready to formulate the LOO stability bounds for RLM.

**Theorem 2.** *Suppose: 1) for each $i \in [n+m]$ and given any $y$, the loss function $\ell(y, f_\theta(X_i))$ is convex and $\rho_i$-Lipschitz in $\theta$; 2) the penalty term $\Omega$ is $\lambda$-strongly convex; 3) for each $i \in [n+m]$, the prediction function $f_\theta(X_i)$ is $\nu_i$-Lipschitz in $\theta$; and 4) given any $y$, the non-conformity score $S(y,z)$ is $\gamma$-Lipschitz in $z$.*[2] *Then, RLM has the following LOO and RO stability bounds.*

$$\tau_{i,j}^{\text{LOO}} = \frac{2\gamma\nu_i(\rho_{n+j} + \bar\rho)}{\lambda(n+1)}, \quad \text{and} \quad \tau_{i,j}^{\text{RO}} = \frac{4\gamma\nu_i\rho_{n+j}}{\lambda(n+1)}, \tag{4}$$

*where $i$ ranges in $[n] \cup \{n+j\}$ for each $1 \leq j \leq m$, and $\bar\rho = n^{-1}\sum_{i=1}^{n}\rho_i$.*

As a side remark, a *uniform* RO stability bound has been established in Ndiaye (2022) (Corollary 3.10). Our RO bound in (4) is non-uniform (not maximizing over $X_i$) and potentially sharper.

### 3.2.2 EXAMPLE 2: STOCHASTIC GRADIENT DESCENT (SGD)

For simplicity, we recap how SGD operates when trained on $\mathcal{D}$. It starts with an initial parameter value $\theta_0$ and runs for $R$ epochs. In each epoch, generate a random permutation $\pi = (\pi_1, \ldots, \pi_n)$ of $[n]$. Then for each $i \in [n]$, update the model parameter by $\theta = \theta - \eta\nabla_\theta\ell(Y_{\pi_i}, f_\theta(X_{\pi_i}))$, where $\eta > 0$ is a user-selected learning rate. After a total of $Rn$ updates, the output $\hat\theta$ is used for prediction. Like in RLM, our LOO stability bound compares two versions of SGD, trained on $\mathcal{D}$ and $\mathcal{D}_j^y$, respectively.

**Theorem 3.** *Suppose: 1) for each $i \in [n+m]$, the loss function $\ell(y, f_\theta(X_i))$ is convex, $\rho_i$-Lipschitz in $\theta$, and its gradient $\nabla_\theta\ell(y, f_\theta(X_i))$ is $\varphi_i$-Lipschitz in $\theta$, for any $y$; 2) for each $i \in [n+m]$, the prediction function $f_\theta(X_i)$ is $\nu_i$-Lipschitz in $\theta$; and 3) the non-conformity score $S(y,z)$ is $\gamma$-Lipschitz in $z$, for any $y$. Then, with learning rate $\eta \leq \frac{2}{\max\{\varphi_i\}}$, SGD has the following LOO and RO stability bounds.*

$$\tau_{i,j}^{\text{LOO}} = R\eta \cdot \gamma\nu_i\rho_{n+j}, \quad \text{and} \quad \tau_{i,j}^{\text{RO}} = 2R\eta \cdot \gamma\nu_i\rho_{n+j}, \tag{5}$$

*where $i$ ranges in $[n] \cup \{n+j\}$ for each $1 \leq j \leq m$.*

Readers may have noticed that for SGD, $\tau_{i,j}^{\text{LOO}}$ is only half of $\tau_{i,j}^{\text{RO}}$, which is very different from the case for RLM (c.f. Theorem 2). The gap here stems from the iterative nature of SGD. Recall that in each epoch, SGD performs $n$ (or $n+1$) gradient descent (GD) updates, with each update depending on a single data point. Consequently, leaving out one data point results in one fewer GD update. In contrast, replacing one data point means performing one GD update differently – in the worst-case scenario, this update may move in opposite directions before and after the replacement, doubling the stability bound.

SGD's iterative nature makes it an excellent example where the number of model fits is the main bottleneck in scaling a crucial learning technique. For SGD, each model fit requires $O(Rn)$ gradient updates, while each prediction costs $O(1)$ time, and evaluating stability bounds for each prediction costs $O(n)$ time. Combining this understanding with Table 1, we see that our method provides significantly faster stable conformal prediction than `RO-StabCP` for performing a large number of predictions.

### 3.2.3 TOWARDS BROADER APPLICABILITY OF LOO-STABCP

**Kernel method:** The kernel method (or "kernel trick") (Schölkopf, 2002) is a commonly used technique in statistical learning. It implicitly transforms data into complex spaces through a kernel function $k(x, x')$. This leads to the reformulated optimization problem: $\hat\theta = \arg\min_{\theta \in \mathbb{R}^n} \frac{1}{n}\sum_{i=1}^{n}\ell(Y_i, k_i^T\theta) + \lambda\theta^T\mathbf{K}\theta$, where $\mathbf{K}$ is a positive-definite kernel matrix $K_{i,j} = k(X_i, X_j)$, and $k_i$ denotes its $i$-th row. It is not difficult to verify that the kernel method is a special case of RLM, thus Theorem 2 applies to the kernel method.

---

[2]Here, $z$ represents the prediction output, and therefore, this is an assumption independent of the model.

**Neural networks:** The stability bounds for RLM and SGD rely on convexity assumptions that might not always hold in practice, such as in (deep) neural networks. Here, we analyze the LOO stability of SGD as a popular optimizer for neural networks, without assuming convexity.

**Theorem 4.** *Assume the conditions of Theorem 3, except that the loss function $\ell(y, f_\theta(X_i))$ is not required to be convex in $\theta$. Then, for the same range of $(i, j)$, SGD has the following LOO and RO stability bounds:*

$$\tau_{i,j}^{\mathrm{LOO}} = R^+ \eta \cdot \gamma \nu_i \rho_{n+j}, \quad and \quad \tau_{i,j}^{\mathrm{RO}} = 2R^+ \eta \cdot \gamma \nu_i \rho_{n+j},$$

*where $R^+ = \sum_{r=1}^{R} \kappa^r$ and $\kappa = \prod_{i=1}^{n}(1 + \eta\varphi_i)$.*

While Theorem 4 does provide a rigorous theoretical justification for neural networks, in practice, the term $\kappa$ may be large if the learning rate $\eta$ is not sufficiently small or the activation function is not very smooth, leading to large Lipschitz constants $\varphi_i$'s. Therefore, this stability bound may turn out to be conservative. Similar to Hardt et al. (2016), we also observed that the empirical stability of SGD for training neural networks is often far better than the worst-case bound described by Theorem 4, see our numerical results in Section 5. This suggests practitioners to still apply the stability bound in Theorem 3, dismissing non-convexity. It is an intriguing but challenging future work to narrow the gap between theory and practice here.

**Bagging:** Bagging (**b**ootstrap **agg**rega**ting**) (Soloff et al., 2024) is a general framework that averages over $B$ models trained on resamples of size $m$ from $\mathcal{D}$. **Random forest** (Wang et al., 2023) is a popular special case of bagging, in which each $f^{(b)}(x)$ is a regression tree. Therefore, we focus on studying bagging. It predicts by $f^B(x) = \frac{1}{B}\sum_{b=1}^{B} f^{(b)}(x)$, where $f^{(b)}(x)$ indicates the individual model trained on the $b$th resample. For simplicity, here we analyze a "derandomized bagging" (Soloff et al., 2024), i.e., setting $B \to \infty$. The prediction function becomes $f^\infty(x) = \mathbb{E}[f^{(b)}(x)]$. Below is the LOO stability of derandomized bagging. Here, we denote $f_j^{y,(b)}(x)$ and $f^{(b)}(x)$ as the individual models obtained from $\mathcal{D}_j^y$ and $\mathcal{D}$, respectively.

**Theorem 5.** *Assume that 1) for any $j \in [m]$ and $(x, y) \in \mathcal{X} \times \mathcal{Y}$, all individual prediction functions $f_j^{y,(b)}(x)$ and $f^{(b)}(x)$ are bounded within a range of width $w_j$; 2) the nonconformity score $S(y, z)$ is $\gamma$-Lipschitz in $z$ for any $y$. Then, derandomized bagging achieves the following LOO stability bound:*

$$\tau_{i,j}^{\mathrm{LOO}} = \frac{\gamma w_j}{2}\sqrt{\frac{p}{1-p}},$$

*where $p = 1 - \left(1 - \frac{1}{n}\right)^m$ and $i \in [n] \cup \{n+j\}$ for $j = 1, \ldots, m$.*

From above, note that the only assumption about the prediction model is bounded output. For example, regression trees satisfy this assumption.

Due to page limit, we relegate more results and discussion to Appendix A.3.

## 4 SIMULATION

In this simulation, we compare several CP methods serving RLM and SGD. We set $n = m = 100$, $\alpha = 0.1$ and generated synthetic data using $X_i \overset{\text{i.i.d.}}{\sim} \mathcal{N}(0, \frac{1}{d}\Sigma)$ with $d = 100$, where $\Sigma_{i,j} = \rho^{|i-j|}$ (i.e., AR(1)). In particular, we chose $\rho = 0.5$ in this experiment. For the response variable we set $Y_i = \mu(X_i; \beta) + \epsilon_i$, where $\epsilon_i \overset{\text{i.i.d.}}{\sim} \mathcal{N}(0, 1)$.

We considered two models for $\mu(\cdot; \beta)$: linear $\mu(x; \beta) = \sum_{j=1}^{d} \beta_j x_j$ and nonlinear $\mu(x; \beta) = \sum_{j=1}^{d} \beta_j e^{x_j/10}$. In both models, set $\beta_j \propto (1 - j/d)^5$ for $j \in [d]$, and normalize: $\|\beta\|_2^2 = d$. To fit the model, we used robust linear regression, equipped with Huber loss:

$$\ell(y, f_\theta(x)) = \begin{cases} \frac{1}{2}(y - f_\theta(x))^2, & \text{if } |y - f_\theta(x)| \leq \epsilon, \\ \epsilon|y - f_\theta(x)| - \frac{1}{2}\epsilon^2, & \text{if } |y - f_\theta(x)| > \epsilon, \end{cases}$$

where $f_\theta(x) = x^T\theta$ and we set $\epsilon = 1$ throughout. We used absolute residual as non-conformity scores. In RLM, we set $\Omega(\theta) = \|\theta\|^2$ and solved it using gradient descent (Diamond & Boyd,

2016). Throughout, we ran SGD for $R = 15$ epochs for all methods, except $R = 5$ for the very slow `FullCP`. For both RLM and SGD, we set the learning rate to be $\eta = 0.001$. For more implementation details, see Appendix G.1. Each experiment was repeated 100 times.

We compare our method to the following benchmarks: 1) `OracleCP` (Appendix C.1, Algorithm 4): an impractical method that uses the true $\{Y_{n+j}\}_{j=1}^{m}$ to predict; 2) `FullCP`; 3) `SplitCP`: 70% training + 30% calibration; and 4) `RO-StabCP`. The performance of each method is evaluated by three measures: 1) coverage probability (method validity); 2) length of predictive interval (prediction accuracy); and 3) computation time (speed).

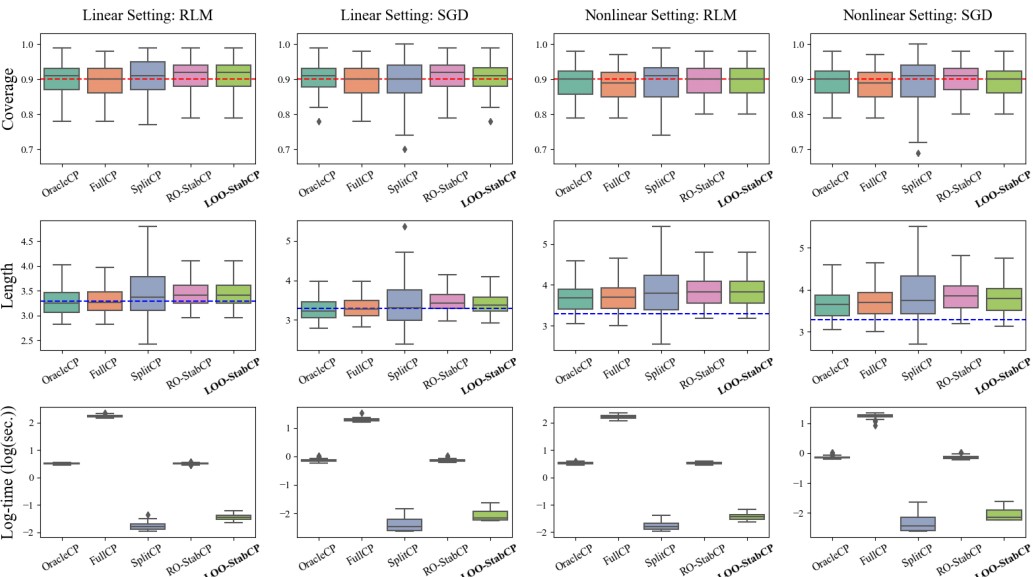

Figure 1: Comparison of CP methods. Our method (`LOO-StabCP`) achieves competitive prediction accuracy and computes at the speed comparable to `SplitCP`, while maintaining coverage validity.

Figure 1 presents the results of our simulation. In the plots for coverage and length, the horizontal dashed lines represent the desired coverage level $(1 - \alpha)$ and the length of the tightest possible prediction band obtained from the true distribution of the data[3], respectively. As expected, all methods maintain valid coverage. Our method shows competitive prediction accuracy, comparable to those of `OracleCP`, `FullCP`, and `RO-StabCP`. These four methods exhibit more consistent and overall superior accuracy compared to `SplitCP`. In terms of computational efficiency, our method performs on par with `SplitCP` and is clearly faster than other methods. Notably, our method significantly outperforms `RO-StabCP` in handling a large number of prediction requests.

## 5 DATA EXAMPLES

We showcase the use of our method on the two real-world data examples analyzed in Ndiaye (2022). The Boston Housing data (Harrison Jr & Rubinfeld, 1978) contain 506 different areas in Boston, each area has 13 features as predictors, such as the *local crime rate* and the *average number of rooms*. The goal is to predict the *median house value* in that area. The Diabetes data (Efron et al., 2004) measured 442 individuals at their "baseline" time points for 10 variables, including *age*, *BMI*, and *blood pressure*, aiming to predict diabetes progression one year after baseline. Both datasets are complete, with no missing entries. All continuous variables have been normalized, and no outliers were identified.

For each data set, we randomly held out $m$ data points (as the test data) for performance evaluation and released the rest to all methods for training/calibration. We tested two settings: $m = 1$ and $m = 100$, two model fitting algorithms: RLM and SGD; and repeated each experiment 100 times.

---

[3]In our setup, it is 3.290 since $\mathbb{P}(|\epsilon_i| \leq 1.645) = 0.9$.

The other configurations, including the model fitted to data ($\ell$, $\epsilon$, $\Omega$, etc.), the list of compared benchmarks and the performance measures, all remained unchanged from Section 4.

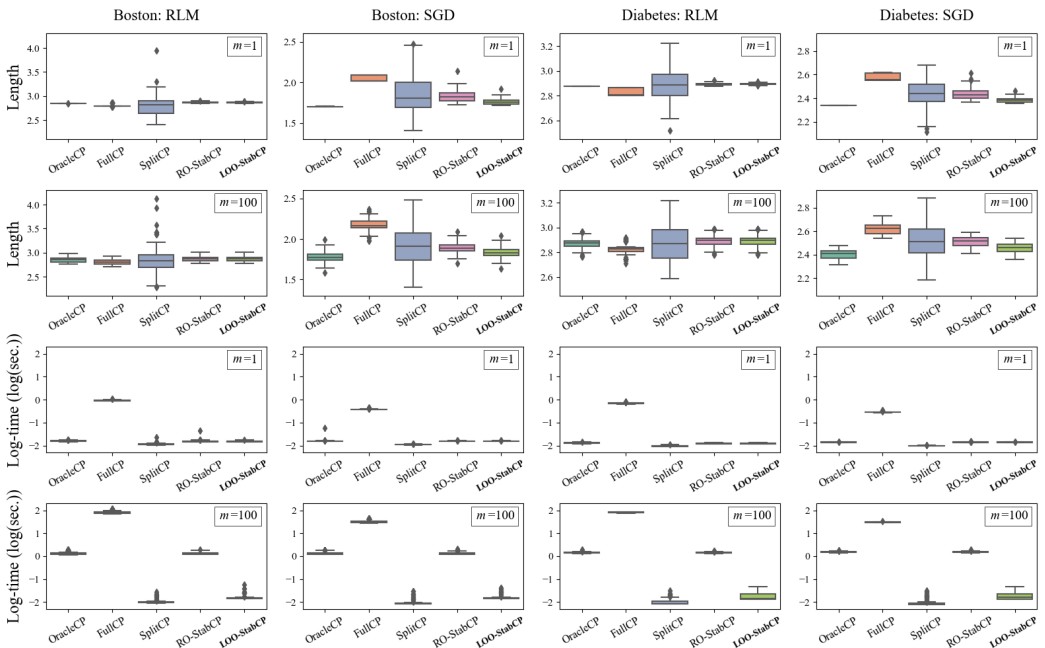

Figure 2: Comparison of prediction interval lengths, under choices of $m = 1$ and $m = 100$.

Figure 2 shows the result. While most interpretations are consistent with that of the simulation, we observe two significant differences between the settings $m = 1$ and $m = 100$. First, under $m = 1$, `RO-StabCP` and our method take comparable time, but when $m$ increases to 100, our method exhibits remarkable speed advantage, as expected. Second, with an increased $m$, the amount of available data for prediction/calibration also decreases. This leads to wider prediction intervals for all methods. Also, `SplitCP` continues to produce more variable and lengthier prediction intervals compared to most other methods for $m \in \{1, 100\}$. In summary, our method `LOO-StabCP` exhibits advantageous performances in all aspects across different settings. The empirical coverage rates are consistent with those in the previous experiments and are provided in Appendix G.3.

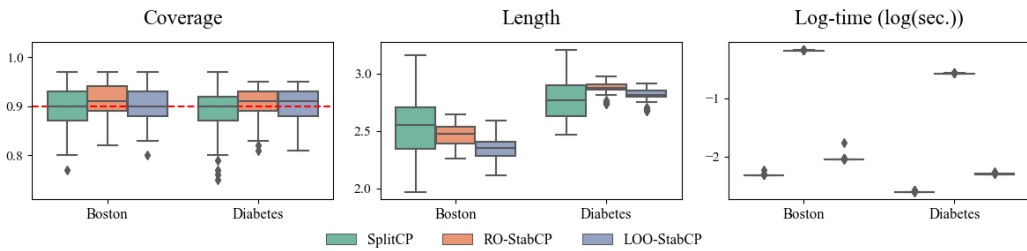

Figure 3: Comparison of CP methods with neural networks with single hidden layer under choice of $m = 100$. `LOO-StabCP` continues to closely achieve the target coverage while exhibiting lower variability in prediction intervals.

To further evaluate the performance of `LOO-StabCP` with non-convex learning methods, we conducted experiments with a neural network of a single hidden layer of 20 nodes and a sigmoid activation function. We set $\eta = 0.001$ and $R = 30$. For stability bounds, we borrowed from the practical guidance in Hardt et al. (2016) and Ndiaye (2022) and set $\tau_{i,j}^{\mathrm{LOO}} \approx R\eta \cdot \gamma \|X_i\| \|X_{n+j}\|$ for `LOO-StabCP` and $\tau_{i,j}^{\mathrm{RO}} \approx 2R\eta \cdot \gamma \|X_i\| \|X_{n+j}\|$ for `RO-StabCP`, respectively. This choice is elaborated in Appendix A.2, see (8). Figure 3 presents the results. `LOO-StabCP` maintained valid

coverage across all scenarios. These findings highlight the robustness of `LOO-StabCP` in handling complex models like neural networks.

Finally, since one could consider derandomization by aggregating results across multiple different splits to reduce variation of SplitCP (Solari & Djordjilović, 2022; Gasparin & Ramdas, 2024), we also numerically compared our method to this approach. The result demonstrated that our `LOO-StabCP` is computationally faster and less conservative than two popular derandomized SplitCP methods Solari & Djordjilović (2022); Gasparin & Ramdas (2024). Due to page limit, we relegate all details of this study to Appendix B.

## 6 APPLICATION: CONFORMALIZED SCREENING

Many decision-making processes, such as drug discovery and hiring, often involve a screening stage to filter among a large number of candidates, prior to more resource-intensive stages like clinical trials and on-site interviews. The data structure is what we have been studying in this paper: training data $\mathcal{D} = \{(X_i, Y_i)\}_{i=1}^n$ and a large number of test points $\{X_{n+j}\}_{j=1}^m$ without observing $Y_{n+j}$'s. Suppose higher values of $Y$ are of interests. Jin & Candès (2023) formulated this as the following *randomized* hypothesis testing problem:

$$H_{0j} : Y_{n+j} \leq c_j \quad \text{vs} \quad H_{1j} : Y_{n+j} > c_j, \quad \text{for } j \in [m],$$

where $c_j$'s are user-selected thresholds (e.g., qualifying score for phone interviews). Then screening candidates means simultaneously testing these $m$ randomized hypotheses. To control for error in multiple testing, one popular criterion is the *false discovery rate* (FDR), defined as the expected false discovery proportion (FDP) among all rejections.

$$\text{FDR} = \mathbb{E}[\text{FDP}], \quad \text{where FDP} = \frac{\sum_{j=1}^m \mathbf{1}\{H_{0j} \text{ is incorrectly rejected}\}}{1 \vee \sum_{j=1}^m \mathbf{1}\{H_{0j} \text{ is rejected}\}}.$$

Jin & Candès (2023) proposed a method called `cfBH` based on `SplitCP`. Our narration will build upon non-conformity scores without repeating details about model fitting. In this context, the non-conformity score should be defined differently, for instance: $S(y, z) = y - z$ without the absolute value, where $y$ is the observed response and $z$ is the fitted value. On the calibration data, $S_i = S(Y_i, \widehat{f}(X_i))$, whereas on the test data, we would consider $S_{n+j}^{c_j} = S(c_j, \widehat{f}(X_{n+j}))$. Jin & Candès (2023)'s `cfBH` method computes the following conformal p-value:

$$p_j^{\text{split}} = \frac{\sum_{i \in \mathcal{I}_{\text{calib}}} \mathbf{1}\{S_i < S_{n+j}^{c_j}\} + 1}{|\mathcal{I}_{\text{calib}}| + 1}, \tag{6}$$

where $\mathcal{I}_{\text{calib}}$ denotes the index set corresponding to the calibration data. To intuitively understand (6), notice that $p_j^{\text{split}} < \alpha$ if and only if $c_j$ falls outside the level-$(1 - \alpha)$ (one-sided) split conformal prediction interval for $Y_{n+j}$. Finally, plugging $\{p_j^{\text{split}}\}_{j=1}^m$ into a Benjamini-Hochberg (BH) procedure (Benjamini & Hochberg, 1995) controls the FDR at a desired level $q$: compute $k^\star = \max\{k : \sum_{j=1}^m \mathbf{1}\{p_j^{\text{split}} \leq qk/m\} \geq k\}$, and reject all $H_{0j}$'s with $p_j^{\text{split}} < qk^\star/m$.

While Jin & Candès (2023)'s method effectively controls FDR and computes fast, the data splitting mechanism leaves space for more thoroughly exploiting available information for model fitting. To this end, we propose a new approach, called `LOO-cfBH` built upon our main method `LOO-StabCP`. We compute stability-adjusted p-values as follows:

$$p_j^{\text{LOO}} = \frac{\sum_{i=1}^n \mathbf{1}\{S_i - \tau_{i,j}^{\text{LOO}} < S_{n+j}^{c_j} + \tau_{n+j,j}^{\text{LOO}}\} + 1}{n + 1}. \tag{7}$$

Algorithm 2 describes the full details of our method.

To numerically compare our method to existing approaches, we used the recruitment data set Ganatara (2020) that was also analyzed in Jin & Candès (2023). It contains 215 individuals, each measured on 12 features such as *education*, *work experience*, and *specialization*. The binary response indicates whether the candidate receives a job offer. We import the robust regression from Section 5 as the prediction method, optimized by SGD. Since the task is classification, we use the

---

**Algorithm 2:** (`LOO-cfBH`) Conformal Selection by Prediction with Leave-One-Out p-values

**Input**   : Training set $\mathcal{D}$, test points $\{X_{n+j}\}_{j=1}^m$, thresholds $\{c_j\}_{j=1}^m$, FDR level $q$.
**Output:** Set of rejected null hypotheses

1. Fit the prediction function $\widehat{f}$ on $\mathcal{D}$;
2. Compute (uncorrected) non-conformity scores on $\mathcal{D}$: $S_i = S(Y_i, \widehat{f}(X_i))$ for $i \in [n]$;
**for** $j \in [m]$ **do**
  | 3. Compute stability bounds $\tau_{i,j}^{\mathrm{LOO}}$ for $i \in [n] \cup \{n+j\}$;
  | 4. Compute LOO conformal p-values $p_j^{\mathrm{LOO}}$ as in (7);
**end**
5. Implement BH Procedure: $k^\star = \max\{k : \sum_{j=1}^m \mathbf{1}\{p_j^{\mathrm{LOO}} \le qk/m\} \ge k\}$ and reject all
  $H_{0j}$'s satisfying $p_j^{\mathrm{LOO}} < qk^\star/m$.

---

clip function in Jin & Candès (2023) as the non-conformity score: $S(y, \widehat{f}) = 100y - \widehat{f}$. For illustration, we also formulated a benchmark `RO-cfBH`, following the spirit of Ndiaye (2022) and using replace-one stability. `RO-cfBH` replaces all $\tau^{\mathrm{LOO}}$ terms in (7) by the corresponding $\tau^{\mathrm{RO}}$ terms; it is otherwise identical to `LOO-cfBH`. We repeated the experiment 1000 times, each time leaving out 20% data points as the test data. In `cfBH`, the data was split into 70% for training and 30% for calibration. We tested three target FDR levels $q \in \{0.1, 0.2, 0.3\}$ and consider three performance measures: 1) FDP; 2) test power, defined as $\left(\sum_{j=1}^m \mathbf{1}\{H_{0j} \text{ is rejected}\}\right) / \left(\sum_{j=1}^m \mathbf{1}\{H_{1j} \text{ is true}\}\right)$; and 3) time cost.

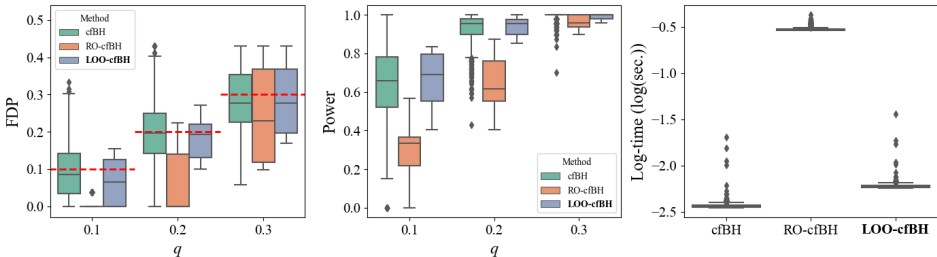

Figure 4: Comparison of screening methods on recruitment data. Time cost does not vary with $q$.

Figure 4 shows the result. Our method achieves valid FDP control for all tested $q$. Compared to `cfBH`, our method is more powerful, due to improved exploitation of available data for prediction. The performance measures also reflect that our method `LOO-cfBH` produces more stable prediction intervals, while sample splitting introduces additional artificial random variations to the result of `cfBH`. Compared to `RO-cfBH`, we highlight our method's significant speed advantage. Moreover, as we showed in Theorem 3, for SGD, our LOO approach achieves a tighter stability bound than RO. As a result, our method is less conservative and more powerful compared to `RO-cfBH`.

# 7   CONCLUSION AND FUTURE WORK

In this paper, we propose a novel approach to stable conformal prediction. Our method significantly improves computational efficiency for multiple prediction requests, compared to the classical stable conformal prediction (Ndiaye, 2022). Here, we mention three directions for future work. First, while we have derived stability bounds for RLM, SGD, neural networks and bagging, improving the tightness of bounds for complex methods remains an important avenue for future research. Second, we have been focusing on continuous responses. It would be an intriguing future work to expand our theory to classification. A third direction is to investigate algorithmic stability for adaptive conformal prediction.

## SUPPLEMENTAL MATERIALS

The Supplemental Materials contains all proofs and additional numerical results. The code for reproducing numerical results is available at: `https://github.com/KiljaeL/LOO-StabCP`.

## ACKNOWLEDGMENT

The authors thank Eugene Ndiaye for helpful discussion and the anonymous reviewers for constructive criticism. This research was supported by NSF grant DMS-2311109.

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

# Supplemental Materials for
# "Leave-one-out Stable Conformal Prediction"

## Kiljae Lee and Yuan Zhang

## A    DETAILED INSIGHTS ON PRACTICAL EXTENSIONS OF LOO-StabCP

In this appendix, we provide further numerical and theoretical analysis to support the approaches discussed in Section 3.2.3.

### A.1    NUMERICAL EXPERIMENTS USING KERNEL TRICK

The key insight of the kernel trick is that by transforming the data into a higher-dimensional feature space using a kernel function, the original optimization problem

$$\widehat{\beta} = \arg\min_{\beta \in \mathbb{R}^d} \frac{1}{n} \sum_{i=1}^{n} \ell(Y_i, X_i^T \beta) + \lambda \|\beta\|^2,$$

can be reformulated as

$$\widehat{\theta} = \arg\min_{\theta \in \mathbb{R}^n} \frac{1}{n} \sum_{i=1}^{n} \ell(Y_i, k_i^T \theta) + \lambda \theta^T \mathbf{K} \theta.$$

This reformulation using the kernel trick does not violate the assumptions required for RLM and SGD, as the transformation maintains the core structure of the optimization problem. Specifically, the kernel matrix $\mathbf{K}$ implicitly defines the high-dimensional feature space through the kernel function $k(X_i, X_j)$, without requiring explicit computation of the transformed features. This ensures that the problem remains computationally tractable.

For RLM, the regularization term $\|\beta\|^2$ in the original formulation translates directly to $\theta^T \mathbf{K} \theta$ in the kernelized version, preserving the strong convexity of the optimization problem as long as we use a positive definite kernel (e.g. radial basis kernel, polynomial kernel, etc.). Similarly, for SGD, the smoothness and Lipschitz continuity of the loss function are preserved, as the transformation affects only the inner product computations, which is linear, and does not alter the fundamental properties of the objective function. Thus, if our original problem satisfies the conditions of LOO stability of RLM and SGD, the kernel trick enables the model to capture nonlinear patterns in the data while ensuring that the theoretical guarantees remain intact.

By integrating the kernel trick, we revisit the scenarios in Section 4, where we initially considered synthetic data examples using standard robust linear regression methods. For our experiments, we employed the radial basis function (RBF) kernel $k_{\mathrm{RBF}}(x, x') = \exp\left(-\frac{\|x-x'\|^2}{2\sigma^2}\right)$ and the polynomial kernel $k_{\mathrm{Poly}}(x, x') = (x^T x' + c)^d$, both chosen for their ability to model complex nonlinear relationships effectively. For hyperparameters, we chose $\sigma = 0.1$, $c = 1$, and $d = 2$. We compared these results to the outcomes in Section 4 and this can be theoretically viewed as a special case of kernel robust regression using a linear kernel.

As shown in Figure 5, `LOO-StabCP` continues to perform reliably under both settings settings without any loss of coverage, validating its adaptability to more sophisticated model structures. Moreover, compared to the linear setting, the use of the kernel trick in nonlinear settings leads to a notable reduction in prediction interval length. This reduction highlights the ability of `LOO-StabCP` with kernel trick to provide more precise predictions while capturing the complex patterns inherent in data, thereby enhancing its practical utility.

### A.2    DETAILED INSIGHTS INTO NONCONVEX OPTIMIZATION

In Section 3.2.3, we derived stability bounds for SGD under nonconvex settings. Here, we provide additional details on the derivation and implications of these bounds.

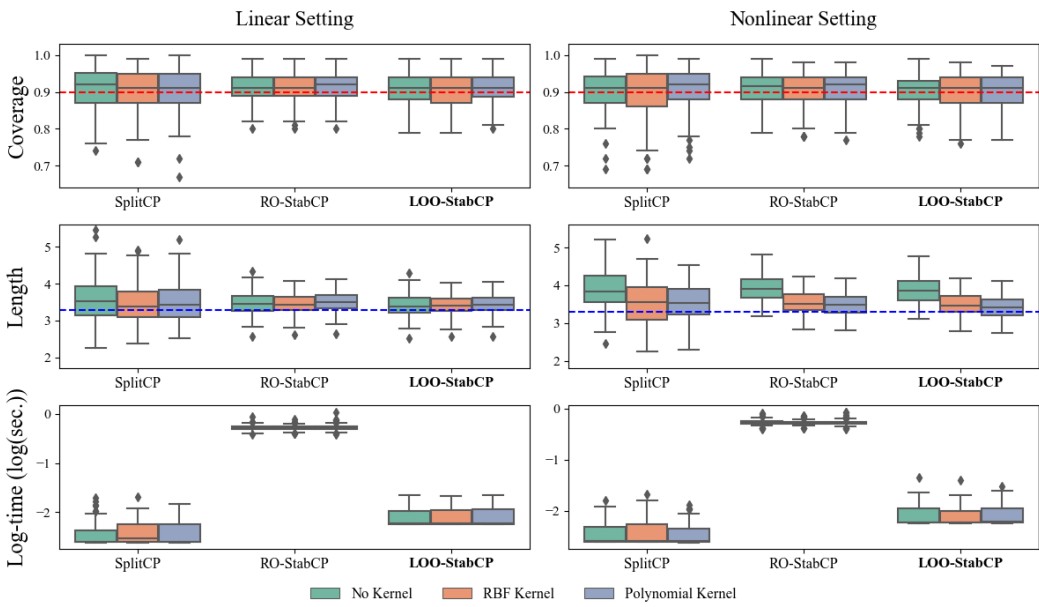

Figure 5: Comparison of CP methods using kernelized technique.

In the convex case (Theorem 3), the stability bounds for SGD are given by:

$$\tau_{i,j}^{\text{LOO}} = R\eta \cdot \gamma\nu_i\rho_{n+j}, \quad \tau_{i,j}^{\text{RO}} = 2R\eta \cdot \gamma\nu_i\rho_{n+j}.$$

On the other hand, for the nonconvex case (Theorem 4), these bounds are modified to include the term $R^+$:

$$\tau_{i,j}^{\text{LOO}} = R^+\eta \cdot \gamma\nu_i\rho_{n+j}, \quad \tau_{i,j}^{\text{RO}} = 2R^+\eta \cdot \gamma\nu_i\rho_{n+j},$$

where $R^+ = \sum_{r=1}^{R} \kappa^r$ and $\kappa = \prod_{i=1}^{n}(1 + \eta\varphi_i)$. Note that the only distinction in the nonconvex case is that $R^+$ replaces $R$. Hence, $R^+$ can be interpreted as representing the cumulative effect of nonconvexity. This term is influenced by the learning rate $\eta$ and the Lipschitz constants of the gradients $\varphi_i$. Specifically, if $\kappa \approx 1$, $R^+$ approximates $R$, aligning with the convex optimization scenario. However, when $\kappa$ is significantly greater than 1, $R^+$ can grow exponentially with $R$, resulting in overly loose bounds. The practical implication of this result is that $\eta$ and $\varphi_i$ significantly influence the tightness of stability bounds. While smaller values of $\eta$ can mitigate this issue, they may also slow down convergence, creating a trade-off between theoretical stability and computational efficiency.

As described in Section 5, we conducted experiments with a neural network featuring a single hidden layer and employed approximated stability bounds:

$$\tau_{i,j}^{\text{LOO}} \approx R\eta \cdot \gamma\|X_i\|\|X_{n+j}\| \quad \text{and} \quad \tau_{i,j}^{\text{RO}} \approx 2R\eta \cdot \gamma\|X_i\|\|X_{n+j}\|. \tag{8}$$

Although these terms approximate our problem as if it were convex, they still capture the interaction between the training and test points in our dataset, providing a practical measure of stability. By using these approximations, we adapted our conformal prediction framework without relying on overly conservative worst-case bounds. Alongside the results in Section 5, Figure 6 shows outcomes for a two-hidden-layer network with 10 and 5 nodes, respectively, under the same settings.

Our empirical results, shown in Figure 3 and Figure 6, demonstrate that these approximations, despite their theoretical looseness, do not compromise the validity of `LOO-StabCP`. These findings are consistent with prior observations (Hardt et al., 2016; Ndiaye, 2022), where theoretical stability bounds in nonconvex settings are often pessimistic, yet empirical results tend to outperform these expectations.

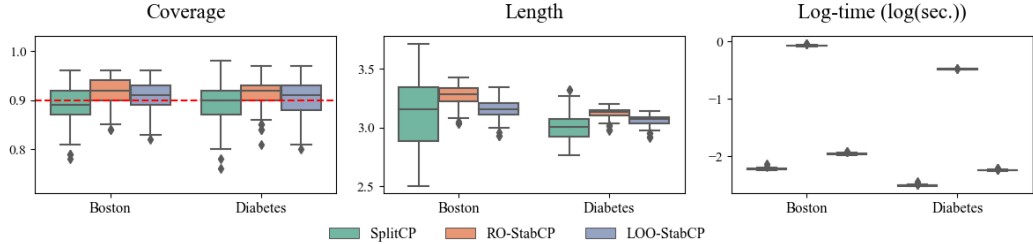

Figure 6: Comparison of CP methods with neural networks with two hidden layers under choice of $m = 100$.

## A.3 ADDITIONAL RESULTS AND DISCUSSION ON BAGGING

While derandomized bagging discussed in 3.2.3 provides conceptual insights into stability, practical bagging methods have internal randomness induced by the resampling scheme. To account for this randomness, here, we provide theoretical results on the LOO stability of bagging in practice.

The randomness in bagging is two-fold. One source is the resampling process, where datasets are created by sampling with replacement from the original dataset. Another arises from the base learning algorithm itself, as seen in random forest, where random feature subsets are selected in each bag. The latter can be characterized by a random variable $\xi \sim \mathcal{U}(0, 1)$. Algorithm 3 illustrates the implementation of bagging.

---

**Algorithm 3:** Bagging

**Input** : Training set $\mathcal{D} = \{(X_i, Y_i)\}_{i=1}^n$, Number of bags $B$, Number of samples in each bag $m$

**Output:** Prediction function $\widehat{f}(\cdot) = f^B(\cdot) := \frac{1}{B} \sum_{b=1}^B f^{(b)}(\cdot)$.

**for** $b \in [B]$ **do**

    1. Sample bag $r^{(b)} = (i_1^{(b)}, \ldots, i_m^{(b)})$ where $i_j^{(b)} \overset{\text{i.i.d.}}{\sim} \mathcal{U}([n])$ for $j \in [m]$;

    2. Sample seed $\xi^{(b)} \sim \mathcal{U}([0, 1])$;

    3. Fit model $f^{(b)}$ with $r^{(b)}$ and $\xi^{(b)}$;

**end**

---

The prediction function of bagging is inherently random, making it challenging to derive a deterministic stability bound. Nonetheless, based on Theorem 5, we can deduce with high probability that bagging is LOO stable. In the context of bagging, $\widehat{f}_j^y(x) = \frac{1}{B} \sum_{b=1}^B f_j^{y,(b)}(x)$ and $\widehat{f}(x) = \frac{1}{B} \sum_{b=1}^B f^{(b)}(x)$, where the sample average replaces the expectation compared to derandomized bagging. The following theorem provides a probabilistic guarantee on the LOO stability bounds for bagging.

**Theorem 6.** *Suppose the conditions of Theorem 5 hold. Then, for any $\delta \in (0, 1)$, bagging has the following LOO stability bounds with probability at least $1 - \delta$.*

$$\tau_{i,j}^{\text{LOO}}(\delta) = \gamma w_j \left\{ \frac{1}{2} \sqrt{\frac{p}{1-p}} + \sqrt{\frac{2}{B} \log\left(\frac{4}{\delta}\right)} \right\}$$

*with $p = 1 - (1 - \frac{1}{n})^m$ where $i$ ranges in $[n] \cup \{n + j\}$ for each $1 \le j \le m$.*

The implications of Theorem 5 and Theorem 6 are as follows. Note that the above theorem requires only the minimal assumption that the base model fitting algorithm used in bagging has bounded output. This suggests that LOO-StabCP can be applied to a wide range of algorithms. For example, building on the stability of bagging, Wang et al. (2023) extended the results to the stability of random forest. Their key insight was that random forest utilize weak decision trees as their base model fitting algorithm, and the final output of a decision tree is always determined as the average of the responses

in the training data it uses. As a result, it is straightforward to see that the output of a decision tree cannot exceed the range of the responses in the training set. Moving forward, these insights can serve as a foundation for exploring stability guarantees in other complex learning algorithms.

# B    COMPARISON WITH DERANDOMIZATION APPROACHES

In this appendix, we extend our numerical experiments to include comparisons with derandomization approaches (Solari & Djordjilović, 2022; Gasparin & Ramdas, 2024), which are potential alternatives to `LOO-StabCP` in terms of reducing the variability of `SplitCP`. Specifically, these methods differ from `SplitCP`, which relies on a single data split, by merging multiple prediction intervals constructed from various splits into one final prediction interval.

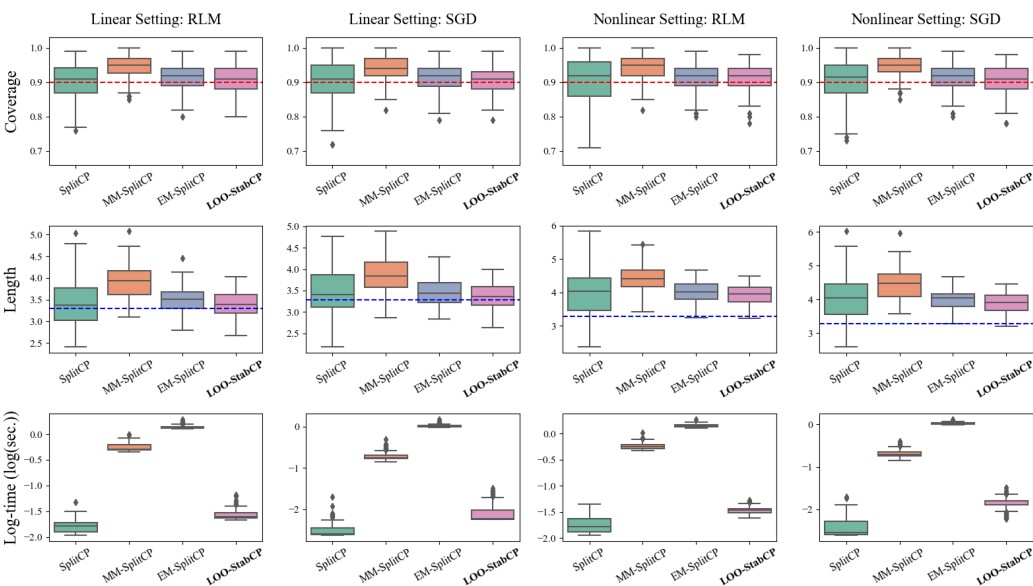

Figure 7: Comparison of CP methods including derandomization approaches on synthetic datasets.

Among these, two notable methods have been proposed in the literature. Solari & Djordjilović (2022) were the first to propose this approach. They generated split conformal prediction intervals with $1 - \frac{\alpha}{2}$ validity from multiple data splits and then derived a final prediction interval through majority voting (i.e., the range covered by more than half of these intervals). They showed that this final interval maintains $1 - \alpha$ validity. We refer to their method as `MM-SplitCP` (Majority Multi-Split Conformal Prediction). Meanwhile, Gasparin & Ramdas (2024) focused on the exchangeability of each prediction interval derived through `MM-SplitCP`. Building on this property, they proposed an alternative aggregation technique that produces tighter yet still valid prediction intervals by applying a majority vote correction. We denote this method as `EM-SplitCP` (Exchangeable Multi-Split Conformal Prediction). For further details on these methods, we refer readers to Solari & Djordjilović (2022); Gasparin & Ramdas (2024).

We compare the performance of these two derandomization techniques with our proposed method, `LOO-StabCP`. To this end, we applied the methods to the settings described in Section 4 and Section 5. For `MM-SplitCP` and `EM-SplitCP`, we merged 30 splits. Figures 7 and 8 present the results on synthetic and real datasets, respectively. From the results, we observe that the variability in coverage and interval length produced by `MM-SplitCP` and `EM-SplitCP` is noticeably lower than that of `SplitCP`, indicating that these derandomization techniques effectively reduce the internal variability of data-splitting approaches.

However, we also find that the average coverage of `MM-SplitCP` and `EM-SplitCP` is generally higher than the predetermined level, suggesting that these derandomization techniques tend to produce conservative intervals. Furthermore, both methods require significantly more computational time, which can be attributed to their reliance on multiple model fits, unlike `SplitCP` and

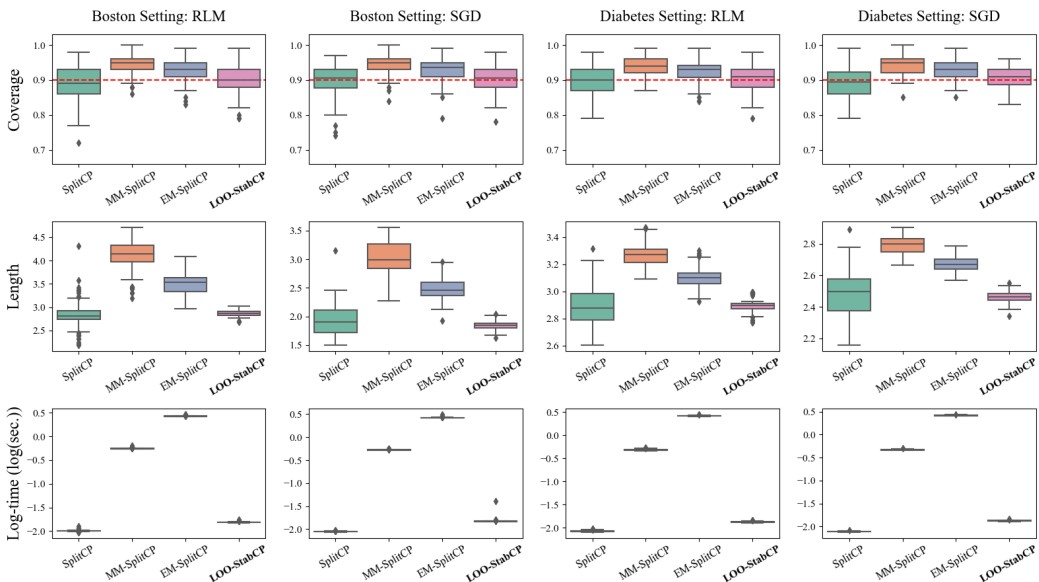

Figure 8: Comparison of CP methods including derandomization approaches on real datasets.

`LOO-StabCP`, which rely on a single model fit. In contrast, `LOO-StabCP` produces tight and stable intervals while maintaining reasonable coverage. These results underscore the computational efficiency and precision of `LOO-StabCP`. These findings are consistent across both synthetic and real data scenarios, showcasing the adaptability and efficiency of `LOO-StabCP` compared to other derandomization methods.

## C  IMPLEMENTATIONS OF CONFORMAL PREDICTION METHODS

### C.1  ORACLE CONFORMAL PREDICTION

---

**Algorithm 4:** (`OracleCP`) Oracle Conformal Prediction Set

**Input**  : Training set $\mathcal{D} = \{(X_i, Y_i)\}_{i=1}^n$, **test sets** $\mathcal{D}_{\text{test}} = \{(X_{n+j}, Y_{n+j})\}_{j=1}^m$, desired coverage $1 - \alpha$.

**Output:** Prediction interval $\mathcal{C}_{j,\alpha}^{\text{oracle}}(X_{n+j})$ for each $j \in [m]$.

**for** $j \in [m]$ **do**
  1. Fit the prediction function $\widehat{f}_j$ on $\mathcal{D}_j^{Y_{n+j}}$;
  2. Compute non-conformity scores on $\mathcal{D}_j^{Y_{n+j}}$: $S_{i,j} = |Y_i - \widehat{f}_j(X_i)|$ for $i \in [n] \cup \{n+j\}$;
  3. Compute prediction interval:
    $\mathcal{C}_{j,\alpha}^{\text{oracle}}(X_{n+j}) = [\widehat{f}_j(X_{n+j}) \pm \mathcal{Q}_{1-\alpha}(\{S_{i,j}\}_{i=1}^n \cup \{S_{n+j,j}\})];$
**end**

---

## C.2 FULL CONFORMAL PREDICTION

---

**Algorithm 5:** (`FullCP`) Full Conformal Prediction Set

---

**Input** : Training set $\mathcal{D} = \{(X_i, Y_i)\}_{i=1}^n$, test points $\{X_{n+j}\}_{j=1}^m$, search grid $\mathcal{G}$, desired coverage $1 - \alpha$.

**Output:** Prediction interval $\mathcal{C}_{j,\alpha}^{\text{full}}(X_{n+j})$ for each $j \in [m]$.

**for** $j \in [m]$ **do**
    **for** $y \in \mathcal{G}$ **do**
        1. Fit the prediction function $\widehat{f}_j^y$ on $\mathcal{D}_j^y$;
        2. Compute non-conformity scores on $\mathcal{D}$: $S_{i,j}^y = |Y_i - \widehat{f}_j^y(X_i)|$ for $i \in [n]$;
        3. Compute quantile value $\mathcal{Q}_{1-\alpha}(\{S_{i,j}^y\}_{i=1}^n \cup \{\infty\})$;
    **end**
    4. Compute prediction interval
    $\mathcal{C}_{j,\alpha}^{\text{full}}(X_{n+j}) = \{y \in \mathcal{G} : S_{n+j,j}^y \leq \mathcal{Q}_{1-\alpha}(\{S_{i,j}^y\}_{i=1}^n \cup \{\infty\})\}$;
**end**

---

## C.3 SPLIT CONFORMAL PREDICTION

---

**Algorithm 6:** (`SplitCP`) Split Conformal Prediction Set

---

**Input** : Training set $\mathcal{D}_{\text{train}} = \{(X_i, Y_i)\}_{i \in \mathcal{I}_{\text{train}}}$, calibration set $\mathcal{D}_{\text{calib}} = \{(X_i, Y_i)\}_{i \in \mathcal{I}_{\text{calib}}}$, test points $\{X_{n+j}\}_{j=1}^m$, desired coverage $1 - \alpha$.

**Output:** Prediction interval $\mathcal{C}_{j,\alpha}^{\text{split}}(X_{n+j})$ for each $j \in [m]$.

1. Fit the prediction function $\widehat{f}_{\text{train}}$ on $\mathcal{D}_{\text{train}}$;
2. Compute non-conformity scores on $\mathcal{D}_{\text{calib}}$: $S_i = |Y_i - \widehat{f}_{\text{train}}(X_i)|$ for $i \in \mathcal{I}_{\text{calib}}$;
**for** $j \in [m]$ **do**
    3. Compute prediction interval
    $\mathcal{C}_{j,\alpha}^{\text{split}}(X_{n+j}) = [\widehat{f}_{\text{train}}(X_{n+j}) \pm \mathcal{Q}_{1-\alpha}(\{S_i\}_{i \in \mathcal{I}_{\text{calib}}} \cup \{\infty\})]$;
**end**

---

## C.4 REPLACE-ONE STABLE CONFORMAL PREDICTION

---

**Algorithm 7:** (`RO-StabCP`) Replace-One Stable Conformal Prediction Set

---

**Input** : Training set $\mathcal{D} = \{(X_i, Y_i)\}_{i=1}^n$, test points $\{X_{n+j}\}_{j=1}^m$, initial guesses $\{\widetilde{y}_{n+j}\}_{j=1}^m$, desired coverage $1 - \alpha$.

**Output:** Prediction interval $\mathcal{C}_{j,\alpha}^{\text{RO}}(X_{n+j})$ for each $j \in [m]$.

**for** $j \in [m]$ **do**
    1. Fit the prediction function $\widehat{f}_j$ on $\mathcal{D}_j^{\widetilde{y}_j}$;
    2. Compute (guessed) non-conformity scores on $\mathcal{D}$: $S_i = |Y_i - \widehat{f}_j(X_i)|$ for $i \in [n]$;
    3. Compute stability bounds $\tau_{i,j}^{\text{RO}}$ for $i \in [n] \cup \{n + j\}$;
    4. Compute prediction interval:
    $\mathcal{C}_{j,\alpha}^{\text{RO}}(X_{n+j}) = [\widehat{f}_j(X_{n+j}) \pm (\mathcal{Q}_{1-\alpha}(\{S_i + \tau_{i,j}^{\text{RO}}\}_{i=1}^n \cup \{\infty\}) + \tau_{n+j,j}^{\text{RO}})]$;
**end**

---

## D  Implementations of LOO Stable Algorithms

### D.1  Regularized Loss Minimization

---

**Algorithm 8:** (RLM) Regularized Loss Minimization

---

**Input** : Training set $\mathcal{D} = \{(X_i, Y_i)\}_{i=1}^n$.
**Output:** Prediction function $\widehat{f}(\cdot) = f_{\widehat{\theta}}(\cdot)$.

1. Compute optimal parameter $\widehat{\theta} := \arg\min_{\theta \in \Theta}\{\frac{1}{n}\sum_{i=1}^n \ell(Y_i, f_\theta(X_i)) + \Omega(\theta)\}$;

---

### D.2  Stochastic Gradient Descent

---

**Algorithm 9:** (SGD) Stochastic Gradient Descent (with Random Reshuffling)

---

**Input** : Training set $\mathcal{D} = \{(X_i, Y_i)\}_{i=1}^n$, number of epochs $R$, step size $\eta$, initial value $\theta_0$.
**Output:** Prediction function $\widehat{f}(\cdot) = f_{\widehat{\theta}}(\cdot)$.

1. Initialize parameter $\theta := \theta_0$;
**for** $r \in [R]$ **do**
    2. Sample a permutation $\pi$ of $[n]$ uniformly at random;
    **for** $i \in [n]$ **do**
        3. Update parameter $\theta := \theta - \eta\nabla_\theta \ell(Y_{\pi_i}, f_\theta(X_{\pi_i}))$;
    **end**
**end**
4. Set the final parameter $\widehat{\theta} := \theta$;

---

## E  Useful Lemmas

**Lemma 1** (Lemma 13.5 in Shalev-Shwartz & Ben-David (2014)). *1. Let $g, h : \mathbb{R}^p \to \mathbb{R}$ be convex function and $\lambda$-strongly convex function, respectively. Then, $g + h$ is $\lambda$-strongly convex. 2. Let $g : \mathbb{R}^p \to \mathbb{R}$ be $\lambda$-strongly convex and $y$ minimize $g$, then, for any $x$,*

$$g(x) - g(y) \geq \frac{\lambda}{2}\|x - y\|^2.$$

**Lemma 2** (Lemma 3.6 in Hardt et al. (2016)). *Let $g : \mathbb{R}^p \to \mathbb{R}$ be a function such that $\nabla g : \mathbb{R}^p \to \mathbb{R}^p$ is a $\varphi$-Lipschitz. Define $h : \mathbb{R}^p \to \mathbb{R}^p$ such that $h(\theta) = \theta - \alpha\nabla g(\theta)$ with $\alpha \leq 2/\varphi$. Then, $h$ is $(1 + \eta\varphi)$-Lipschitz. If $g$ is in addition convex, $h$ is 1-Lipschitz.*

## F  Proofs of Theorems

### F.1  Proof of Theorem 1

*Proof.* By Definition 2, we have $S_{i,j}^y \leq S_i + \tau_{i,j}^{\mathrm{LOO}}$, for $i \in [n]$ and $j = [m]$. Similarly, for any $j$, we have $|y - \widehat{f}(X_{n+j})| - \tau_{n+j,j}^{\mathrm{LOO}} \leq S_{n+j,j}^y$. Therefore, for any $j$, the following holds for all $y$ contained in $\mathcal{C}_{j,\alpha}^{\mathrm{full}}(X_{n+j})$:

$$|y - \widehat{f}(X_{n+j})| - \tau_{n+j,j}^{\mathrm{LOO}} \leq \mathcal{Q}_{1-\alpha}(\{S_i + \tau_{i,j}^{\mathrm{LOO}}\}_{i=1}^n \cup \{\infty\}),$$

which is equivalent to

$$y \in [\widehat{f}(X_{n+j}) \pm (\mathcal{Q}_{1-\alpha}(\{S_i + \tau_{i,j}^{\mathrm{LOO}}\}_{i=1}^n \cup \{\infty\}) + \tau_{n+j,j}^{\mathrm{LOO}})].$$

This directly implies $\mathcal{C}_{j,\alpha}^{\mathrm{LOO}}(X_{n+j}) \supseteq \mathcal{C}_{j,\alpha}^{\mathrm{full}}(X_{n+j})$ and hence

$$\mathbb{P}(Y_{n+j} \in \mathcal{C}_{j,\alpha}^{\mathrm{LOO}}(X_{n+j})) \geq \mathbb{P}(Y_{n+j} \in \mathcal{C}_{j,\alpha}^{\mathrm{full}}(X_{n+j})) \geq 1 - \alpha,$$

for any choice of $\alpha$.

$\square$

### F.2 PROOF OF THEOREM 2

*Proof.* For $y \in \mathcal{Y}$ and $j \in [m]$ define $F_j^y(\theta) = \frac{1}{n+1}\{\sum_{i=1}^n \ell(Y_i, f_\theta(X_i)) + \ell(y, f_\theta(X_{n+j}))\} + \Omega(\theta)$ and $F(\theta) = \frac{1}{n}\{\sum_{i=1}^n \ell(Y_i, f_\theta(X_i))\} + \Omega(\theta)$. Then, $\widehat{\theta}_j^y = \arg\min_{\theta \in \Theta} F_j^y(\theta)$ and $\widehat{\theta} = \arg\min_{\theta \in \Theta} F(\theta)$.

We begin by proving the LOO algorithmic stability. Fix $y$ and $j$ and suppose that $\|\widehat{\theta}_j^y - \widehat{\theta}\| \leq \frac{2(\rho_{n+j}+\bar\rho)}{\lambda(n+1)}$. Then, for all $i \in [n] \cup \{n+j\}$ and $z \in \mathcal{Y}$,

$$
\begin{aligned}
|S(z, \widehat{f}_j^y(X_i)) - S(z, \widehat{f}(X_i))| &\leq \gamma|\widehat{f}_j^y(X_i) - \widehat{f}(X_i)| \\
&= \gamma|f_{\widehat{\theta}_j^y}(X_i) - f_{\widehat{\theta}}(X_i)| \\
&\leq \gamma\nu_i\|\widehat{\theta}_j^y - \widehat{\theta}\| \\
&\leq \frac{2\gamma\nu_i(\rho_{n+j}+\bar\rho)}{\lambda(n+1)}.
\end{aligned}
$$

The first and the second inequalities follow from the Lipschitz property of non-conformity score function and the prediction function, respectively. Therefore, it suffices to obtain the bound of $\|\widehat{\theta}^y - \widehat{\theta}\|$ as assumed above. By the first part of Lemma 1, $F_j^y$ and $F$ are $\lambda$-strongly convex functions of $\theta$. Using the second part of the Lemma 1, we have:

$$
\begin{aligned}
\frac{\lambda}{2}\|\widehat{\theta} - \widehat{\theta}_j^y\|^2 &\leq F_j^y(\widehat{\theta}) - F_j^y(\widehat{\theta}_j^y) \\
&= \frac{n}{n+1}\{F(\widehat{\theta}) - F(\widehat{\theta}_j^y)\} \\
&\quad + \frac{1}{n+1}\{\ell(y, f_{\widehat{\theta}}(X_{n+j})) - \ell(y, f_{\widehat{\theta}_j^y}(X_{n+j})) + \Omega(\widehat{\theta}) - \Omega(\widehat{\theta}_j^y)\} \\
&\leq \frac{1}{n+1}\{\ell(y, f_{\widehat{\theta}}(X_{n+j})) - \ell(y, f_{\widehat{\theta}_j^y}(X_{n+j})) + \Omega(\widehat{\theta}) - \Omega(\widehat{\theta}_j^y)\}.
\end{aligned}
\tag{9}
$$

The last inequality follows from the optimality of $\widehat{\theta}$. Now, by the Lipschitz property of the loss function, we have:

$$
\ell(y, f_{\widehat{\theta}}(X_{n+j})) - \ell(y, f_{\widehat{\theta}_j^y}(X_{n+j})) \leq \rho_{n+j}\|\widehat{\theta} - \widehat{\theta}_j^y\|.
\tag{10}
$$

On the other hand, again by the optimality of $\widehat{\theta}$, it holds that

$$
0 \leq F(\widehat{\theta}_j^y) - F(\widehat{\theta}) = \frac{1}{n}\sum_{i=1}^n \{\ell(Y_i, f_{\widehat{\theta}_j^y}(X_i)) - \ell(Y_i, f_{\widehat{\theta}}(X_i)) + \Omega(\widehat{\theta}^y) - \Omega(\widehat{\theta}),
$$

which implies

$$
\Omega(\widehat{\theta}) - \Omega(\widehat{\theta}_j^y) \leq \bar\rho\|\widehat{\theta}_j^y - \widehat{\theta}\|,
\tag{11}
$$

by the Lipschitz property of the loss function. Finally, combining (10) and (11) to (9), we get $\|\widehat{\theta}_j^y - \widehat{\theta}\| \leq \frac{2(\rho_{n+j}+\bar\rho)}{\lambda(n+1)}$.

For the RO algorithmic stability, fix $y$, $\widetilde{y}$, and $j$. By the similar arguments as for (9), we have

$$
\begin{aligned}
\frac{\lambda}{2}\|\widehat{\theta}_j^{\widetilde{y}} - \widehat{\theta}_j^y\|^2 &\leq \frac{1}{n+1}\{\ell(y, f_{\widehat{\theta}_j^{\widetilde{y}}}(X_{n+j})) - \ell(y, f_{\widehat{\theta}_j^y}(X_{n+j}))\} \\
&\quad + \frac{1}{n+1}\{\ell(\widetilde{y}, f_{\widehat{\theta}_j^y}(X_{n+j})) - \ell(\widetilde{y}, f_{\widehat{\theta}_j^{\widetilde{y}}}(X_{n+j}))\} \\
&\leq \frac{2\rho_{n+j}}{n+1}\|\widehat{\theta}_j^y - \widehat{\theta}_j^{\widetilde{y}}\|,
\end{aligned}
$$

and this implies $\|\widehat{\theta}_j^y - \widehat{\theta}_j^{\widetilde{y}}\| \leq \frac{4\rho_{n+j}}{\lambda(n+1)}$. The rest of the proof is similar to the previous case.

$\square$

### F.3 PROOF OF THEOREM 3

*Proof.* We start with proving the case of RO algorithmic stability first, for clearer presentation. Also, we only prove the case of $j = 1$ and $R = 1$ since extending to the case of $j > 1$ or $R > 1$ is straightforward. Let $\pi$ be an arbitrary permutation of $[n+1]$ and $k$ be such that $\pi_k = n+1$. Fix $y$ and $\widetilde{y}$. Let $(\theta_0^y, \theta_1^y, \ldots, \theta_{n+1}^y)$ and $(\theta_0^{\widetilde{y}}, \theta_1^{\widetilde{y}}, \ldots, \theta_{n+1}^{\widetilde{y}})$ be the updating sequences of SGD sharing $\pi$ for $\mathcal{D}_j^y$ and $\mathcal{D}_j^{\widetilde{y}}$, respectively. Note that $\widehat{f}_j^y = \widehat{f}_1^y = f_{\theta_{n+1}^y}$ and $\widehat{f}_j^{\widetilde{y}} = \widehat{f}_1^{\widetilde{y}} = f_{\theta_{n+1}^{\widetilde{y}}}$. As in the proof of Theorem 2, we first bound the distance between the two terminal parameters, $\|\theta_{n+1}^y - \theta_{n+1}^{\widetilde{y}}\|$.

Let us first consider the case of $k = n+1$. Then, by the SGD update rule, we can see that $\theta_i^y = \theta_i^{\widetilde{y}}$ for all $i = [n]$ since for SGD update, the two sequences share the first $n$ data points as well as the initial parameter. Therefore, we have

$$
\begin{aligned}
\|\theta_{n+1}^y - \theta_{n+1}^{\widetilde{y}}\| &= \|\{\theta_n^y - \eta\nabla_\theta \ell(y, f_{\theta_n^y}(X_{n+1}))\} - \{\theta_n^{\widetilde{y}} - \eta\nabla_\theta \ell(\widetilde{y}, f_{\theta_n^{\widetilde{y}}}(X_{n+1}))\}\| \\
&\leq \eta\|\nabla_\theta \ell(y, f_{\theta_n^y}(X_{n+1}))\| + \eta\|\nabla_\theta \ell(\widetilde{y}, f_{\theta_n^{\widetilde{y}}}(X_{n+1}))\| \\
&\leq 2\eta\rho_{n+1}.
\end{aligned}
$$

Here, we used triangle inequality, and then the Lipschitz property of loss function.

Now, consider the case of $k < n+1$. If $i = k$, by the similar argument, we can show that $\|\theta_i^y - \theta_i^{\widetilde{y}}\| \leq \|\theta_{i-1}^y - \theta_{i-1}^{\widetilde{y}}\| + 2\eta\rho_{n+1}$. Otherwise, if $i \neq k$, by Lemma 2 with the choice of $\alpha := \eta$, $\varphi := \varphi_{\pi_i} \leq \frac{2}{\eta}$, and $g(\theta) := \ell(Y_{\pi_i}, f_\theta(X_{\pi_i}))$, we have:

$$
\begin{aligned}
\|\theta_i^y - \theta_i^{\widetilde{y}}\| &= \|\{\theta_{i-1}^y - \eta\nabla_\theta \ell(Y_{\pi_i}, f_{\theta_i^y}(X_{\pi_i}))\} - \{\theta_{i-1}^{\widetilde{y}} - \eta\nabla_\theta \ell(Y_{\pi_i}, f_{\theta_i^{\widetilde{y}}}(X_{\pi_i}))\}\| \\
&\leq \|\theta_i^y - \theta_i^{\widetilde{y}}\|,
\end{aligned}
\tag{12}
$$

since $\ell(Y_{\pi_i}, f_\theta(X_{\pi_i}))$ is convex. Unraveling the recursion from the above two inequality, we get $\|\theta_{n+1}^y - \theta_{n+1}^{\widetilde{y}}\| \leq \|\theta_0^y - \theta_0^{\widetilde{y}}\| + 2\eta\rho_{n+1} = 2\eta\rho_{n+1}$. The last equality holds since the two updating sequences share the common initial value. The remaining parts follow similarly to the proof of Theorem 2.

Next, to prove the LOO algorithmic stability, fix $y$ and let $\pi' = (\pi_1', \ldots, \pi_n')$ be the sequence obtained from $\pi$ by excluding the $k$th entry. For example, if we choose $n = 4$ and $\pi = (3, 2, 5, 1, 4)$, then $k = 3$ and $\pi' = (3, 2, 1, 4)$. Then, it can be shown that $\pi'$ is an arbitrary permutation of $[n]$. Define an updating sequence of SGD, $(\theta_0, \theta_1, \ldots, \theta_n)$ for $\mathcal{D}$ induced by $\pi'$, i.e, $\widehat{f} = f_{\theta_n}$. Note that $\theta_0 = \theta_0^y$. As the case of the RO algorithmic stability, it suffices to show that $\|\theta_{n+1}^y - \theta_n\| \leq \eta\rho_{n+1}$.

If $k = n+1$, then we have $\theta_i^y = \theta_i$ for $i = [n]$. Therefore, it follows that

$$
\begin{aligned}
\|\theta_{n+1}^y - \theta_n\| &= \|\{\theta_n^y - \eta\nabla_\theta \ell(y, f_{\theta_n^y}(X_{n+1}))\} - \theta_n\| \\
&\leq \eta\|\nabla_\theta \ell(y, f_{\theta_n^y}(X_{n+1}))\| \\
&\leq \eta\rho_{n+j}.
\end{aligned}
$$

For the case of $k < n+1$ and further remaining parts, we can follow the same procedure used in the RO algorithmic stability. $\qquad\square$

### F.4 PROOF OF THEOREM 4

*Proof.* The overall structure of the proof is almost identical to that of Theorem 3. Again, let us focus on the proof of RO stability with $R = 1$ first. Recall that in that proof, the convexity assumption was used only in (12). Since we have discarded the convexity assumption of $\ell(Y_{\pi_i}, f_\theta(X_{\pi_i}))$ by Lemma 2 again, the Lipschitz constant of $h(\theta) = \theta - \eta\nabla_\theta \ell(Y_{\pi_i}, f_\theta(X_{\pi_i}))$ is replaced from 1 to $1 + \eta\varphi_{\pi_i}$. That is, we obtain the following recursive inequalities:

$$
\|\theta_i^y - \theta_i^{\widetilde{y}}\| \leq \begin{cases} \|\theta_{i-1}^y - \theta_{i-1}^{\widetilde{y}}\| + 2\eta\rho_{n+1} & \text{if} \quad i = k, \\ (1 + \eta\varphi_{\pi_i})\|\theta_{i-1}^y - \theta_{i-1}^{\widetilde{y}}\| & \text{if} \quad i < k. \end{cases}
\tag{13}
$$

Considering $\|\theta_0^y - \theta_0^{\widetilde{y}}\| = 0$, unraveling these inequalities yields

$$\|\theta_{n+1}^y - \theta_{n+1}^{\widetilde{y}}\| \leq \left( \prod_{i=k+1}^{n+1} (1 + \eta\varphi_{\pi_i}) \right) \cdot 2\eta\rho_{n+j}$$

$$\leq \left( \prod_{i=1}^{n+1} (1 + \eta\varphi_{\pi_i}) \right) \cdot 2\eta\rho_{n+j}$$

$$= \left( \prod_{i=1}^{n+1} (1 + \eta\varphi_i) \right) \cdot 2\eta\rho_{n+j}$$

$$= 2\kappa\eta\rho_{n+j},$$

since $\eta\varphi_i > 0$ by definitions. Extending this to the case of $R > 1$ is not as straightforward as the proof of Theorem 3, hence we also present the corresponding proof. In this case, we can use induction. Set $R > 1$ and let $r \in [R]$. Suppose that up to $(r-1)$th epoch, the difference of parameter is bounded by $2\left( \sum_{s=1}^{r-1} \kappa^s \right) \eta\rho_{n+j}$. Then, $r$th iteration can be treated as the case of $R = 1$ with $\|\theta_0^y - \theta_0^{\widetilde{y}}\| \leq 2\left( \sum_{s=1}^{r-1} \kappa^s \right) \eta\rho_{n+j}$. In this case, unraveling (13) yileds

$$\|\theta_{n+1}^y - \theta_{n+1}^{\widetilde{y}}\| \leq \left( \prod_{i=k+1}^{n+1} (1 + \eta\varphi_{\pi_i}) \right) \left[ \left( \prod_{i=1}^{k-1} (1 + \eta\varphi_{\pi_i}) \right) \|\theta_0^y - \theta_0^{\widetilde{y}}\| + 2\eta\rho_{n+j} \right]$$

$$\leq \left( \prod_{i=k+1}^{n+1} (1 + \eta\varphi_{\pi_i}) \right) \left[ \left( \prod_{i=1}^{k-1} (1 + \eta\varphi_{\pi_i}) \right) 2 \left( \sum_{s=1}^{r-1} \kappa^s \right) \eta\rho_{n+j} + 2\eta\rho_{n+j} \right]$$

$$\leq 2\eta\rho_{n+j} \left[ \left( \prod_{i \neq k} (1 + \eta\varphi_{\pi_i}) \right) \left( \sum_{s=1}^{r-1} \kappa^s \right) + \left( \prod_{i=k+1}^{n+1} (1 + \eta\varphi_{\pi_i}) \right) \right]$$

$$\leq 2\kappa\eta\rho_{n+j} \left[ \left( \sum_{s=1}^{r-1} \kappa^s \right) + 1 \right]$$

$$= 2 \left( \sum_{s=1}^{r} \kappa^s \right) \eta\rho_{n+j}.$$

Since we already proved the case of $r = 1$, this completes proof for RO stability. For the LOO stability, we can use the same reasoning. □

### F.5 Proof of Theorem 5

*Proof.* Fix $(x, y) \in \mathcal{X} \times \mathcal{Y}$ and $j \in [m]$. Due to the symmetry of the resampling scheme, i.e., sampling uniformly with replacement, we have

$$\widehat{f}(x) = \mathbb{E}\left[ f_j^{y,(b)}(x) \middle| n + 1 \notin r \right].$$

Therefore, using the above facts along with the Lipschitz property of the non-conformity score function, we get

$$|S(z, \widehat{f}_j^y(x)) - S(z, \widehat{f}(x))| \leq \gamma \left| \widehat{f}_j^y(x) - \widehat{f}(x) \right|$$

$$= \gamma \left| \mathbb{E}\left[ f_j^{y,(b)}(x) \right] - \mathbb{E}\left[ f_j^{y,(b)}(x) \middle| n + 1 \notin r \right] \right|$$

$$= \gamma \left| \mathbb{E}\left[ f_j^{y,(b)}(x) - \mathbb{E}\left[ f_j^{y,(b)}(x) \right] \middle| n + 1 \notin r \right] \right|.$$

Next, by the definitions of conditional expectation and covariance,

$$\mathbb{E}\left[ f_j^{y,(b)}(x) - \mathbb{E}\left[ f_j^{y,(b)}(x) \right] \middle| n + 1 \notin r \right] = \frac{1}{\mathbb{P}(n + 1 \notin r)} \mathbb{E}\left[ \left\{ f_j^{y,(b)}(x) - \mathbb{E}\left[ f_j^{y,(b)}(x) \right] \right\} \mathbf{1}\{n + 1 \notin r\} \right]$$

$$= \frac{1}{\mathbb{P}(n + 1 \notin r)} \text{Cov}\left( f_j^{y,(b)}(x), \mathbf{1}\{n + 1 \notin r\} \right).$$

Combining the above results, we have

$$|S(z, \widehat{f}_j^y(x)) - S(z, \widehat{f}(x))| \le \frac{1}{1-p} \left| \mathrm{Cov}\left( f_j^{y,(b)}(x), \mathbf{1}\{n+1 \notin r\} \right) \right|, \tag{14}$$

where $p = \mathbb{P}(n+1 \in r) = 1 - (1 - \frac{1}{n})^m$. Furthermore, it holds that

$$\left| \mathrm{Cov}\left( f_j^{y,(b)}(x), \mathbf{1}\{n+1 \notin r\} \right) \right| \le \left[ \mathrm{Var}\left( f_j^{y,(b)}(x) \right) \mathrm{Var}\left( \mathbf{1}\{n+1 \notin r\} \right) \right]^{\frac{1}{2}}$$

$$\le \sqrt{\frac{w_j^2}{4} p(1-p)}$$

$$= \frac{w_j}{2} \sqrt{p(1-p)}.$$

Here, the first inequality follows from the Cauchy-Schwarz inequality. For the second inequality, we apply Popoviciu's inequality for variance and the properties of the Bernoulli distribution. Substituting this bound into (14) completes the proof. □

### F.6 Proof of Theorem 6

*Proof.* Fix $(x, y) \in \mathcal{X} \times \mathcal{Y}$ and $j \in [m]$. Let $\widehat{f}_j^y(x)$ and $\widehat{f}(x)$ denote the predictions corresponding to bagging, and let $\widehat{f}_j^{y,\infty}(x)$ and $\widehat{f}^\infty(x)$ denote the predictions corresponding to derandomized bagging. Then,

$$|S(z, \widehat{f}_j^y(x)) - S(z, \widehat{f}(x))| \le \gamma \left| \widehat{f}_j^y(x) - \widehat{f}(x) \right|$$

$$= \gamma \left| \left\{ \widehat{f}_j^y(x) - \widehat{f}_j^{y,\infty}(x) \right\} + \left\{ \widehat{f}_j^{y,\infty}(x) - \widehat{f}^\infty(x) \right\} + \left\{ \widehat{f}^\infty(x) - \widehat{f}(x) \right\} \right|$$

$$\le \gamma \left[ \left| \widehat{f}_j^y(x) - \widehat{f}_j^{y,\infty}(x) \right| + \left| \widehat{f}_j^{y,\infty}(x) - \widehat{f}^\infty(x) \right| + \left| \widehat{f}^\infty(x) - \widehat{f}(x) \right| \right]. \tag{15}$$

Consider each term on the last line of (15). For the first term, note that

$$\left| \widehat{f}_j^y(x) - \widehat{f}_j^{y,\infty}(x) \right| = \left| \frac{1}{B} \sum_{b=1}^B f_j^{y,(b)}(x) - \mathbb{E}\left[ f_j^{y,(b)}(x) \right] \right|.$$

Since each single prediction $f_j^{y,(b)}(x)$ is almost surely bounded within an interval of range $w_j$, by Hoeffding's inequality, we have

$$\mathbb{P}\left( \left| \widehat{f}_j^y(x) - \widehat{f}_j^{y,\infty}(x) \right| \le t \right) \ge 1 - 2\exp\left( -2Bt^2/w_j^2 \right),$$

for any $t > 0$. Setting $\frac{\delta}{2} = 2\exp(-2Bt^2/w_j^2)$ yields

$$\mathbb{P}\left( \left| \widehat{f}_j^y(x) - \widehat{f}_j^{y,\infty}(x) \right| \le \sqrt{\frac{w_j^2}{2B} \log\left( \frac{4}{\delta} \right)} \right) \ge 1 - \frac{\delta}{2}.$$

Similarly, for the third term, we obtain an identical bound:

$$\mathbb{P}\left( \left| \widehat{f}(x) - \widehat{f}^\infty(x) \right| \le \sqrt{\frac{w_j^2}{2B} \log\left( \frac{4}{\delta} \right)} \right) \ge 1 - \frac{\delta}{2}.$$

For the second term, a direct application of Theorem 5 gives the following deterministic bound:

$$\left| \widehat{f}_j^{y,\infty}(x) - \widehat{f}^\infty(x) \right| \le \frac{\gamma w_j}{2} \sqrt{\frac{p}{1-p}}.$$

Combining all the bounds for the three terms in (15) using the union bound, we have that, with probability at least $1 - \delta$,

$$|S(z, \widehat{f}_j^y(x)) - S(z, \widehat{f}(x))| \leq \gamma \left[ \sqrt{\frac{w_j^2}{2B} \log\left(\frac{4}{\delta}\right)} + \frac{w_j}{2} \sqrt{\frac{p}{1-p}} + \sqrt{\frac{w_j^2}{2B} \log\left(\frac{4}{\delta}\right)} \right]$$

$$= \gamma w_j \left\{ \frac{1}{2} \sqrt{\frac{p}{1-p}} + \sqrt{\frac{2}{B} \log\left(\frac{4}{\delta}\right)} \right\}.$$

$\square$

## G  DETAILS OF NUMERICAL EXPERIMENTS

### G.1  DETAILS OF ALGORITHMS

The configurations satisfy the assumptions of Theorem 2 and Theorem 3, allowing us to compute the stability bounds concretely. First, for RLM, the following stability bounds were used. For $i \in [n] \cup \{n + j\}$ for each $j \in [m]$,

$$\tau_{i,j}^{\text{LOO}} = \frac{2\epsilon\|X_i\|}{\lambda(n+1)} \left( \|X_{n+j}\| + \frac{1}{n} \sum_{i=1}^{n} \|X_i\| \right), \quad \tau_{i,j}^{\text{RO}} = \frac{4\epsilon\|X_i\|\|X_{n+j}\|}{\lambda(n+1)}.$$

Next, the stability bounds for SGD are as follows:

$$\tau_{i,j}^{\text{LOO}} = R\eta\epsilon\|X_i\|\|X_{n+j}\|, \quad \tau_{i,j}^{\text{RO}} = 2R\eta\epsilon\|X_i\|\|X_{n+j}\|.$$

### G.2  ADDITIONAL RESULTS FROM SECTION 4

| | | | OracleCP | FullCP | SplitCP | RO-StabCP | **LOO-StabCP** |
|---|---|---|---|---|---|---|---|
| Linear | RLM | Coverage | 0.903 (0.040) | 0.896 (0.043) | 0.903 (0.060) | 0.910 (0.039) | 0.910 (0.039) |
| | | Length | 3.272 (0.250) | 3.300 (0.257) | 3.455 (0.514) | 3.442 (0.257) | 3.442 (0.257) |
| | | Time | 3.201 (0.172) | 176.783 (15.704) | 0.017 (0.006) | 3.190 (0.176) | 0.035 (0.008) |
| | SGD | Coverage | 0.903 (0.041) | 0.896 (0.043) | 0.900 (0.059) | 0.911 (0.040) | 0.906 (0.040) |
| | | Length | 3.252 (0.250) | 3.300 (0.257) | 3.420 (0.557) | 3.464 (0.259) | 3.405 (0.259) |
| | | Time | 0.720 (0.087) | 19.320 (2.018) | 0.005 (0.003) | 0.720 (0.075) | 0.009 (0.005) |
| Nonlinear | RLM | Coverage | 0.892 (0.045) | 0.886 (0.047) | 0.893 (0.059) | 0.897 (0.044) | 0.897 (0.044) |
| | | Length | 3.659 (0.317) | 3.690 (0.340) | 3.812 (0.554) | 3.828 (0.344) | 3.827 (0.344) |
| | | Time | 3.289 (0.219) | 163.101 (22.120) | 0.017 (0.005) | 3.275 (0.221) | 0.038 (0.010) |
| | SGD | Coverage | 0.892 (0.045) | 0.886 (0.047) | 0.895 (0.062) | 0.900 (0.043) | 0.894 (0.044) |
| | | Length | 3.641 (0.318) | 3.690 (0.340) | 3.855 (0.612) | 3.849 (0.345) | 3.789 (0.345) |
| | | Time | 0.732 (0.074) | 17.425 (2.206) | 0.005 (0.003) | 0.746 (0.093) | 0.009 (0.005) |

Table 2: Mean (and standard deviation) of empirical coverage, average prediction interval length, and execution time across 100 iterations for each scenario in simulation.

## G.3 ADDITIONAL RESULTS FROM SECTION 5

| | | | OracleCP | FullCP | SplitCP | RO-StabCP | **LOO-StabCP** |
|---|---|---|---|---|---|---|---|
| $m = 1$ | Boston | RLM | 0.920 (0.273) | 0.910 (0.288) | 0.910 (0.288) | 0.920 (0.273) | 0.920 (0.273) |
| | | SGD | 0.900 (0.302) | 0.920 (0.273) | 0.910 (0.288) | 0.900 (0.302) | 0.900 (0.302) |
| | Diabetes | RLM | 0.910 (0.288) | 0.900 (0.302) | 0.890 (0.314) | 0.910 (0.288) | 0.910 (0.288) |
| | | SGD | 0.920 (0.273) | 0.910 (0.288) | 0.920 (0.273) | 0.930 (0.256) | 0.920 (0.273) |
| $m = 100$ | Boston | RLM | 0.906 (0.031) | 0.897 (0.031) | 0.898 (0.040) | 0.905 (0.030) | 0.905 (0.030) |
| | | SGD | 0.905 (0.029) | 0.901 (0.032) | 0.901 (0.036) | 0.910 (0.028) | 0.906 (0.029) |
| | Diabetes | RLM | 0.900 (0.035) | 0.889 (0.037) | 0.894 (0.043) | 0.900 (0.035) | 0.900 (0.035) |
| | | SGD | 0.902 (0.031) | 0.890 (0.036) | 0.902 (0.037) | 0.914 (0.030) | 0.906 (0.031) |

Table 3: The mean (and the standard deviation) of empirical coverage over 100 iterations for each scenario on Boston Housing and Diabetes datasets.

## G.4 ADDITIONAL RESULTS FROM SECTION 6

| | | cfBH | RO-cfBH | **LOO-cfBH** |
|---|---|---|---|---|
| $q = 0.1$ | FDP | 0.0928 (0.0713) | 0.0038 (0.0115) | 0.0657 (0.0617) |
| | Power | 0.6319 (0.2053) | 0.3041 (0.1452) | 0.6744 (0.1295) |
| | Time | 0.0037 (0.0006) | 0.2976 (0.0114) | 0.0060 (0.0011) |
| $q = 0.2$ | FDP | 0.2000 (0.0807) | 0.0602 (0.0813) | 0.1836 (0.0560) |
| | Power | 0.9277 (0.0838) | 0.6522 (0.1450) | 0.9430 (0.0486) |
| | Time | 0.0037 (0.0005) | 0.2971 (0.0100) | 0.0060 (0.0002) |
| $q = 0.3$ | FDP | 0.2882 (0.0806) | 0.2483 (0.1212) | 0.2837 (0.0934) |
| | Power | 0.9923 (0.0198) | 0.9627 (0.0347) | 0.9917 (0.0136) |
| | Time | 0.0037 (0.0002) | 0.2970 (0.0101) | 0.0060 (0.0003) |

Table 4: Mean (and standard deviation) of FDP, power, and execution time for three conformal selection methods.

