# OpenReview forum: "Leave-One-Out Stable Conformal Prediction"
_ICLR.cc/2025/Conference — ICLR 2025 Poster_

### Official Review · Reviewer_VLB9 · 2024-10-17

**Soundness:** 3
**Presentation:** 3
**Contribution:** 2
**Rating:** 5
**Confidence:** 4

**Summary:**

This paper proposes a leave-one-out algorithm stability definition, which the authors utilize to reduce the computational burden of the full conformal prediction method. The finite-sample validity of the prediction interval is proved. The authors have provided some experimental results to show the superiority of the proposed method in computation speed.

**Strengths:**

The proposed method does reduce the computational burden by circumventing model refitting when computing prediction intervals for distinct test points. The computational complexity is then reduced by $m$ times. This is also validated by the numerical experiments.

**Weaknesses:**

The method proposed by the authors is quite similar to the baseline established by Ndiaye (2022). Given this foundation, the extension presented by the authors appears somewhat straightforward, leading to a limited contribution. While the proposed method significantly reduces computational burden, it relies on the assumption that the learning algorithm adheres to the LOO stability assumption. Since conformal prediction is elegant for its wide applicability as a wrapper around any machine learning model, the examples RLM and SGD discussed in the paper are very limited. This limitation is particularly notable in the current landscape, where sophisticated deep learning models and LLMs are prevalent.

**Questions:**

1. The only issue I care about is whether commonly used lightweight machine learning algorithms like the random forest or simple neural networks can satisfy the LOO stability proposed in the paper. It will be a major issue and I will be willing to raise my score with extra theoretical results.

---

> ### Author Response · Authors · 2024-11-23
>
> Thank you for your comments.
> We understand that your main concern regards novelty, as **you commented**:
>
> *“The method proposed by the authors is quite similar to the baseline established by Ndiaye (2022). Given this foundation, the extension presented by the authors appears somewhat straightforward, leading to a limited contribution.”*
>
> **Response:**
>
> We would like to clarify several important points, as follows:
>
> **First**, while our work builds on the series of work by Lei and Wasserman (2014), Barber et al. (2021), and Ndiaye (2022), our LOO-StabCP method presents a novel stability mechanism that parallels Ndiaye's RO-StabCP. The field of stable conformal prediction naturally follows a common narration flow: first propose a stability type, then establish theoretical guarantees, and finally analyze a few examples/applications, accompanied by numerical studies. However, the key innovations lie in the specific stability mechanisms and their resulting theoretical and computational characteristics. Our theoretical results (e.g., compare the bounds in Theorem 2) and numerical studies demonstrate that our proposed mechanism has significant advantages.
>
> We intentionally maintained a parallel structure to Ndiaye (2022) for an easier comparison of results, especially for readers familiar with that work. This similarity in the order of presenting scientific contents is non-essential and should not diminish the novelty of our work because the scientific contents themselves are original. Our method achieves significant improvements in both computational efficiency and accuracy for multiple prediction problems and therefore should not be considered a very minor variation of Ndiaye (2022).
>
> **Second**, we emphasize that the leave-one-out stability, compared to replace-one, represents a non-obvious, even a bit counter-intuitive, methodological innovation. Recall that both Ndiaye (2022) and our work are variations of full conformal. Our work challenges the doctrine in full conformal prediction that including the $(n+1)$th data point in model fitting is required. Even Ndiaye (2022) still follows this rule. Beginners to conformal prediction are usually warned by their teachers that training only on data points $1,\ldots,n$ would be a mistake for full conformal. In this regard, our leave-one-out stability hits a conceptual blind spot. Like many methodological innovations, the simplicity of our approach becomes more apparent in hindsight (than in foresight).
>
> **Third**, our proposal is driven by strong practical need and effectively addresses a key issue in Ndiaye's RO-StabCP: computational inefficiency in handling many prediction requests. With the recent development in combining FDR control with conformal prediction, computational efficiency of the conformal prediction method has become a spotlight issue for distribution-free statistical inference (Jin \& Candes, 2022).
>
> In the revised paper, motivated by the comments of all reviewers, we also expanded the applications/examples much beyond Ndiaye (2022). This also represents significantly nontrivial advancement. See the color-highlighted contents in the revised paper, including the appendices.
>
> ---
>
> **You also commented**:
>
> *“... Since conformal prediction is elegant for its wide applicability as a wrapper around any machine learning model, the examples RLM and SGD discussed in the paper are very limited. This limitation is particularly notable in the current landscape, where sophisticated deep learning models and LLMs are prevalent.”*
>
> and
>
> *“The only issue I care about is whether commonly used lightweight machine learning algorithms like the random forest or simple neural networks can satisfy the LOO stability proposed in the paper. It will be a major issue and I will be willing to raise my score with extra theoretical results.”*
>
> **Response:**
>
> Thank you for spurring us to explore more applications of our method.
> We understand that these points are closely related to the common comment made by most reviewers on whether our method is applicable to more examples, including (deep) neural networks. Therefore, we address these two comments in our general reply to all reviewers. Please refer to bullet point 2 for our reply to your comment questioning the apparent limitation of our applications, and other bullet points there for our new results applying our method to neural networks via SGD (Theorem 4 in the revised manuscript).

---

> > ### Author Response · Authors · 2024-11-23
> > **References:**
> >
> > [1] Jing Lei and Larry Wasserman. Distribution-free prediction bands for non-parametric regression. *Journal of the Royal Statistical Society: Series B (Statistical Methodology)*, 76(1):71–96, 2014.
> >
> > [2] Rina Foygel Barber, Emmanuel J. Candès, Aaditya Ramdas, and Ryan J. Tibshirani. Predictive inference with the jackknife+. *The Annals of Statistics*, 49(1):486–507, 2021. doi: 10.1214/20-AOS1965. https://doi.org/10.1214/20-AOS1965.
> >
> > [3] Eugene Ndiaye. Stable conformal prediction sets. In *International Conference on Machine Learning*, pp. 16462–16479. PMLR, 2022.
> >
> > [4] Ying Jin and Emmanuel J Candès. Selection by prediction with conformal p-values. *Journal of Machine Learning Research*, 24(244):1–41, 2023.

---

> ### Comment · Reviewer_VLB9 · 2024-11-24
>
> I appreciate the authors' efforts to improve the article. While I remain unconvinced by the clarification regarding the novelty of the work, I acknowledge the effort put into providing additional theoretical results. In light of this, I am willing to raise my score to 5.

---

> > ### Author Response · Authors · 2024-11-24
> >
> > Thank you for raising the score.  We completely understand that different readers may have different opinions about a given paper, as other reviewers find our paper's contributions novel.  We appreciate your comments which encouraged us to explore further applications of our method.  Time is limited (since the reviews posted), but we will further work on directions pointed out by you and other reviewers in the future.  Thank you again.

---

### Official Review · Reviewer_LXUa · 2024-11-02

**Soundness:** 3
**Presentation:** 3
**Contribution:** 3
**Rating:** 8
**Confidence:** 3

**Summary:**

This paper introduces Leave-One-Out Stable Conformal Prediction (LOO-StabCP), a novel conformal prediction approach that enhances computational efficiency for multiple predictions while ensuring coverage guarantees. LOO-StabCP builds on prior work by Ndiaye (2022), which utilized replace-one stability, by leveraging leave-one-out stability to require only a single model fit on the training data, regardless of the number of test points. This eliminates the need for computationally intensive refits for each individual test point. The authors validate LOO-StabCP with theoretical stability bounds for regularized loss minimization (RLM) and stochastic gradient descent (SGD)--methods widely used in modern machine learning. Extensive experiments on synthetic and real-world datasets show that LOO-StabCP matches or exceeds existing methods in predictive accuracy, efficiency, and computational speed.

**Strengths:**

The paper makes a contribution to conformal prediction by addressing a critical need for efficient uncertainty quantification in large-scale applications, especially when predictions are needed across multiple test points. Building on prior work that used algorithmic stability for conformal prediction, LOO-StabCP introduces a novel approach to leave-one-out stability by treating the model trained on the dataset (=training dataset $\mathcal{D}$) as one generated by “leaving out” each test point from an augmented set (as noted in Section 3.1 following Algorithm 1). This refinement results in a significant speedup, especially valuable in scenarios with numerous predictions.

The authors provide a solid theoretical foundation to support their proposed LOO-StabCP by deriving stability bounds for Regularized Loss Minimization (RLM) and Stochastic Gradient Descent (SGD), strengthening the method’s rigor. Additionally, empirical results on both synthetic and real-world datasets effectively demonstrate LOO-StabCP’s advantages in computational speed, prediction interval size and statistical power. The comparison of computational complexity across methods (e.g., FullCP, SplitCP, RO-StabCP), summarized in Table 1, clearly showcases the efficiency gains, and the illustration of practical applications in screening tasks, described in Section 6, highlights the method’s utility.

**Weaknesses:**

Overall, I believe this is a well-written paper.  However, some refinements in presentation could potentially enhance clarity and completeness, even though it is already quite solid.  Here are some specific suggestions.

**1. Adding Remarks & Prospects for Broader Examples/Applications:**
Adding comments on the tightness of stability bounds for the RLM and SGD examples in Section 3.2 would improve the interpretability of results, and clarify the sharpness or looseness of the current results.
Additionally, while RLM and SGD are valuable examples, discussing potential extensions to other models or methods, along with conjectures or insights about the scope of the applications, would help readers better understand the broader applicability of the proposed approach. Furthermore, this is a minor point, but for improved parallelism between Sections 3.2.1 and 3.2.2, it may be worth noting that RLM and SGD are not strictly parallel choices: RLM represents a problem formulation, whereas SGD is an optimization algorithm used to solve such problems.

**2. Augmenting Experiments:**
As one way to o assess the tightness of the proposed LOO-StabCP prediction intervals, the authors could compare the intervals obtained by LOO-StabCP against the tightest possible prediction set with coverage $1-\alpha$ as a benchmark, which can be calculated in numerical experiments, for instance, using the $\alpha/2$- and $(1-\alpha/2)$-quantiles of instantiated predictions from multiple runs.  This comparison would provide further insights into the tightness of the proposed method (as well as other CP methods being considered in the study).
Also, broadening the simulation studies to include a wider range of model types would offer greater assurance that LOO-StabCP's performance gains are not specific to the tested settings. Expanding these results, perhaps in an Appendix, would help readers evaluate the method’s effectiveness across diverse applications.

**3. Ensuring Notation Consistency and Simplification:**
Ensuring that all notation is defined before use (e.g., $[1:n]$) would improve accessibility. Simplifying notation where possible--for instance, by fixing  $j=1$ in Section 2  to reduce complexity without sacrificing rigor--could enhance readability. While the authors may have intended to highlight the dependence on the test point index $j$, simplifying this notation in Section 2 and then generalizing back in Section 3 could make the initial section more approachable.

**Questions:**

Here is a list of questions and minor suggestions.

- *Line 34:* I think $Y_{n+j}$ should not be included in $\mathcal{D}_{\textrm{test}}$.
- *Line 58:* Consider using “RO-StabCP” for clarity instead of “in Ndiaye (2022),” which is already referenced in the preceding paragraph.
- *Line 70:* The phrase “guess $Y_{n+j}$ with $y$” may not read clearly.
- *Line 75:* Specifying the range, e.g., by “…swapped for $i, i' \in [n] \cup \{n+j\}$” would be clearer.
- *Line 90:* Typo: $y$ should be replaced by $i$.
- *Line 103 (Definition 1):* (1) Define $[1:m]$ notation; (2) clarify the quantifier regarding $\mathcal{D}$.
- *Line 108:* “Recall” may be clearer than “Let.”
- *Line 140:* Consider adding remarks right after Definition 2 to discuss (1) the pursuit of adaptive parameters rather than uniform bounds to obtain sharper stability estimates, (2) the practicality of assuming known parameters, and (3) how these parameters impact the accuracy and robustness of prediction intervals. Although these points are addressed in later sections, readers would benefit from a brief mention here.
- *Line 154:* Using varied parenthesis sizes or brackets might improve readability.
- *Line 202:* Recall the meaning of the augmented data $\mathcal{D}^y_j$ from Line 70 for context.
- *Line 367:* The statement “This leads to wider prediction intervals for all methods, and particularly for SplitCP, more variability in prediction interval length” is not clear to me. It suggests the prediction interval of SplitCP becomes particularly wider due to the increase $m=1 \to m=100$.  However, SplitCP already appears to vary similarly at both $m=1$ and $m=100$, while the other methods vary more at $m=100$.
- *Line 369:* While the authors note that derandomization (Gasparin \& Ramdas, 2024) would incur extra computational costs, how does LOO-StabCP compare with derandomization in other aspects such as prediction accuracy, coverage, and stability?
- *Line 465:* The comment “Compared to cfBH, our method is more powerful” could benefit from further clarification. How is this conclusion drawn from Figure 3 and Table 3?

---

> ### Author Response · Authors · 2024-11-23
>
> Thank you so much for reading our paper very carefully and for the numerous insightful comments. Here is our point-by-point response to your comments.
>
> ## Main comments
>
> ---
>
> 1. **Adding Remarks & Prospects for Broader Examples/Applications**
>
> Thank you for this comment. We understand that this is a common concern shared by other reviewers. Therefore, we address it in the general reply to all reviewers.
>
> ---
>
> 2. **Augmenting Experiments**
>
> We sincerely thank you for your insightful suggestion to assess the tightness of the proposed LOO-StabCP prediction intervals by comparing them to the theoretically tightest possible prediction intervals under the true data distribution. This is indeed an excellent point that has enhanced the interpretability of our results. In particular, in our simulation study, where the data distribution is known, the tightest possible interval for all predictions corresponds to twice the $(1-\frac{\alpha}{2})$-quantile of the standard normal distribution.
>
> To address your suggestion, we incorporated this information into our results by adding horizontal dashed lines to the plots for interval length, similar to how desired coverage is presented in coverage plots. See Figures 1, 5, and 7. This inclusion provided valuable insights. For instance, in Figure 1, which uses a linear model, the results closely approach the tightest interval under the linear data-generating process (DGP). However, in nonlinear scenarios, model misspecification leads to wider intervals for all CP methods, as expected. Furthermore, in Figure 5, we observe that the addition of kernel methods under the nonlinear setting brings the intervals closer to the conceptual tightest width, demonstrating their effectiveness in mitigating model misspecification. We are grateful for your suggestion, which not only improved our analysis but also provided a deeper understanding of the performance of LOO-StabCP and other CP methods in various scenarios.
>
> You also asked us to test our method's performance in more settings. In the revision, we added numerical studies for the following application scenarios:
>    - Kernel method (see Figure 5)
>    - One- and two-layer neural networks (see Figures 3 and 6)
>    - Comparison with derandomized SplitCP, in response to the comments by you and Reviewer Rbu1 (Appendix Section B)
>
> ---
>
> 3. **Ensuring Notation Consistency and Simplification**
>
> Thank you for this comment.
>
> In fact, this was exactly a point that we were torn and spent a lot of time discussing when writing up this paper. Exactly as you mentioned, "what (the non-conformity score) $S$ depends on" is important. The reason why our method can speed up over RO-StabCP is because it depends on less things. Pressure also came from the strict page limit, compared to the amount of contents we wish to present.
>
> We completely agree with you that it is desirable to reduce notation and make symbols as light as possible, especially for the introduction part. Following your advice, we have abbreviated $S_{i,j}^y$ as $S_i^y$ for the full and split conformal, as well as suppressed the dependency of the prediction set on $j$ there.
> Meanwhile, please allow us to explain the several-fold difficulties we faced and considerations.
>
> On the dependency of $S$ on $i$: as you know, we want to show the source of the heavier computational burden of RO-StabCP, therefore, we explicitly wrote down the dependency of its non-conformity scores on $j$. But RO-StabCP builds upon full conformal, in the review of full conformal, we might want to avoid disconnection of notation (full conformal's notation is simplified, while RO-StabCP is not). Also, in SplitCP, most non-conformity scores do not depend on $y$ and $j$. Abbreviating the notation throughout Section 2 might make this difference less obvious. On the dependency of $S$ on $y$: we explicitly wrote the dependency of $S$ on $y$ to simplify the justification of full conformal, RO-StabCP and LOO-StabCP.
>
> If we write Section 2 without the symbol $j$, then at the beginning of Section 3, we worry about the space it would cost to properly restate the previous results, especially for RO-StabCP, adding index $j$ back. Our manuscript, which is currently not completely doing this, is already at the 10 page limit. Finally, our understanding is that our readers might tend to already have some exposure to conformal prediction and won't use our paper as the entry point to this area.
>
> With these considerations, we simplified the notation for full and split conformal reviews, while still keeping the full notation for RO-StabCP. We added a heads-up for readers at the beginning of RO-StabCP: *"From now on, we will switch back to the full notation for $S$ and no longer abbreviate $S_{i,j}^y$ as $S_i^y$."*
>
> We would appreciate your further feedback on the clarity of the new version's presentation of Section 2, as well as any advice from you on how to reduce notation without causing the aforementioned potential issues. Thank you so much!

---

> ### Author Response · Authors · 2024-11-23
>
> ## Other questions and minor suggestions
>
> - **Line 34:** *I think $Y_{n+j}$ should not be included in $\mathcal{D}_{\text{test}}$.*
>   Thank you. In the revision, we have emphasized that $Y_{n+j}$ is unobserved and now write "$Y_{n+j}=?$" instead of just $Y_{n+j}$ in $\mathcal{D}_{\text{test}}$.
>
> - **Line 58:** *Consider using "RO-StabCP" for clarity instead of "in Ndiaye (2022)."*
>   Thank you, done.
>
> - **Line 70:** *The phrase "guess $Y_{n+j}$ with '$y$'" may not read clearly.*
>   Sure, we have replaced it with "*let $y$ denote a guessed value of the unobserved $Y\_{n+j}$.*"
>
> - **Line 75:** *Specifying the range, e.g., by "...swapped for $i,i'\in[n]\cup \{n+j\}$" would be clearer.*
>   Thank you, done.
>
> - **Line 90:** *Typo: $y$ should be replaced by $i$.*
>   Thank you, done.
>
> - **Line 103 (Definition 1):** *(1) Define $[1:m]$ notation; (2) clarify the quantifier regarding $\mathcal{D}$.*
>   Thank you for the suggestion.
>   (1) We have switched to the notation "$[m]$" as you suggested (globally replaced all).
>   (2) In the revised Definition 1, we have clarified that "*$\hat{f}\_j^{\mathfrak{y}}$ is trained on $\mathcal{D}\cup \\{(X_{n+j}, \mathfrak{y})\\}$, for $\mathfrak{y}=y$ or $\tilde{y}$.*"
>
> - **Line 108:** *"Recall" may be clearer than "Let."*
>
>   Thank you, done.
>
> - **Line 140:** *Consider adding remarks right after Definition 2 to discuss (1) the pursuit of adaptive parameters rather than uniform bounds to obtain sharper stability estimates, (2) the practicality of assuming known parameters, and (3) how these parameters impact the accuracy and robustness of prediction intervals. Although these points are addressed in later sections, readers would benefit from a brief mention here.*
>
>   Thank you for this insightful suggestion.  We have revised the paragraph immediately following Definition 2 to address your concerns. Please refer to the texts there highlighted in blue.
>
> - **Line 154:** *Using varied parenthesis sizes or brackets might improve readability.*
>
>   Thank you. We have varied a pair of round parentheses to curly brackets. We also adjusted the sizes of some parentheses/brackets. Please let us know if the current version reads better.
>
> - **Line 202:** *Recall the meaning of the augmented data ${\cal D}_j^y$ from Line 70 for context.*
>
>   Thank you, done.
>
> - **Line 367:** *The statement "This leads to wider prediction intervals for all methods, and particularly for SplitCP, more variability in prediction interval length" is not clear to me. It suggests the prediction interval of SplitCP becomes particularly wider due to the increase $m=1\to m=100$. However, SplitCP already appears to vary similarly at both $m=1$ and $m=100$, while the other methods vary more at $m=100$.*
>
>   Thank you for pointing this out. You're right that this sentence shouldn't be comparing $m=1$ versus $m=100$. In the revision, we have replaced this sentence with a statement that marginally compares SplitCP with other methods for $m=1$ and $m=100$, respectively.
>
> - **Line 369:** *While the authors note that derandomization (Gasparin \& Ramdas, 2024) would incur extra computational costs, how does LOO-StabCP compare with derandomization in other aspects such as prediction accuracy, coverage, and stability?*
>
>   Thank you for this suggestion. Reviewer Rbu1 also mentioned this point. In this revision, we added a new Appendix B to compare our method to derandomized approaches. The conclusion is that our method generally computes faster and suffers less conservatism.
>
> - **Line 465:** *The comment "Compared to cfBH, our method is more powerful" could benefit from further clarification. How is this conclusion drawn from Figure 3 and Table 3?*
>
>   Thank you for your thoughtful question. First, we clarify that Table 3, redundant to Figure 3, has been moved to the appendix, and Figure 3 has been renumbered as Figure 4 in the revised version. In Figure 4, we compare LOO-cfBH and cfBH across three FDR levels ($q = 0.1, 0.2, 0.3$). At lower $q$ levels, LOO-cfBH achieves lower FDP and higher power than cfBH. This empirical finding indicates that LOO-cfBH is able to correctly reject a greater proportion of false null hypotheses with a smaller proportion of false rejections, highlighting its ability to perform more precise screening compared to cfBH. We hope this clarifies our findings.

---

> > ### Comment · Reviewer_LXUa · 2024-11-26
> >
> > I appreciate the authors' efforts to address the questions and comments raised during the review process. The revisions and responses have satisfactorily addressed my concerns and questions, and hence, I maintain my positive rating for the paper.

---

> > > ### Author Response · Authors · 2024-11-26
> > >
> > > We sincerely thank you for maintaining your positive opinion of our paper! Addressing the various thoughtful and insightful concerns you raised allowed us to delve deeper into our work, and we are confident that it has significantly helped us develop our discussion to a more profound level. Once again, we truly appreciate your invaluable support.

---

### Official Review · Reviewer_Rbu1 · 2024-11-04

**Soundness:** 3
**Presentation:** 3
**Contribution:** 2
**Rating:** 6
**Confidence:** 4

**Summary:**

The paper proposes Leave-One-Out Stable Conformal Prediction (LOO-StabCP) to speed up full conformal using algorithmic stability without sample splitting for better balance of computational efficiency and prediction accuracy. This method is much faster in handling a large number of prediction requests compared to existing method RO-StabCP based on replace-one stability. The authors show that their method is theoretically justified and demonstrates superior numerical performance on synthetic and real-world data.

**Strengths:**

* The paper proposes a novel method to address the problem in conformal prediction lies in balancing computation cost with prediction accuracy.
* Numerical results in this paper show that the proposed method achieves a competitive average coverage and a higher power compared to existing methods.

**Weaknesses:**

* The authors introduce leave-one-out algorithmic stability for stability correction. However, it is difficult to calculate the stability bounds $\tau_{i,j}^{\mathrm{LOO}}$ when a complex deep learning algorithm is chosen to fit the model.

* When trying to derive the LOO stability bound of RLM and SGD described in section 3.2, similar problems are encountered as mentioned above: the conditions in Theorems 2 \& 3 are difficult to verify.

* Numerical experiments are insufficient: the covariates of synthetic data are set to be independent; the prediction algorithms used are all robust linear regression. More complex settings and algorithms should be considered and compared.

**Questions:**

* I think it is necessary to discuss more and carefully about applying LOO-StabCP to deep learning algorithms for reasons outlined in the weaknesses box.

* As introduced in section 1, derandomization methods can address the issue of decreasing accuracy causing by randomly split but increase computational cost. I think it is better to demonstrate the performance of some derandomization method and compare it with LOO-StabCP to illustate the differences between the two in all aspects.

* The authors compute stability-adjusted p-values for multiple selection in section 6. I think it is better to verify the validity of p-values in (7) and state it as a proposotion for completeness.

---

> ### Author Response · Authors · 2024-11-23
>
> Thank you very much for your valuable suggestions. Here is a point-by-point response to your concerns.
>
>
> ## Your comments:
>
> ---
>
> 1. **Comment:**
>    "*The authors introduce leave-one-out algorithmic stability for stability correction. However, it is difficult to calculate the stability bounds $\tau_{i,j}^\mathrm{LOO}$ when a complex deep learning algorithm is chosen to fit the model.*"
>
> 2. **Comment:**
>    "*When trying to derive the LOO stability bound of RLM and SGD described in section 3.2, similar problems are encountered as mentioned above: the conditions in Theorems 2 \& 3 are difficult to verify.*"
>
> **Response:**
> Thank you for your insightful comments 1 \& 2. We address them together.
>
> While neural networks are structurally complex, they are built of linear functions and activation functions, combined through composition. Using the chain rule, we can compute stability bounds by calculating derivatives layer by layer, as long as we choose an activation function with Lipschitz derivatives—common nonlinear activation functions, such as the sigmoid function or its scaled variant, the hyperbolic tangent, are Lipschitz. This process enables us to determine the Lipschitz constants for each layer and combine them to obtain the overall Lipschitz constant for the network.
>
> Alternatively, we can also use practical approximations to these stability bounds. For instance, rough estimates of the stability terms can be calculated by analyzing interactions between training and test data points in the feature space (see Appendix A.2 for more details). These approximations provide a feasible way to assess stability without requiring precise constants. As demonstrated in our numerical experiments (Figure 5), even with these approximations, LOO-StabCP maintained valid coverage and achieved competitive prediction interval tightness. This highlights the practical feasibility of applying LOO-StabCP to neural networks.
>
> ---
>
> 3. **Comment:**
>    "*Numerical experiments are insufficient: the covariates of synthetic data are set to be independent; the prediction algorithms used are all robust linear regression. More complex settings and algorithms should be considered and compared.*"
>
> **Response:**
> We agree that the original experiments relied on simplified settings, such as independent covariates and the use of robust linear regression, which might not fully demonstrate the generality of our method. To address this, we have significantly expanded the scope of our numerical experiments in the revised manuscript to include more complex settings and algorithms.
>
> First, we introduced dependencies in the synthetic data by incorporating an AR(1) covariance structure for the covariates, with a correlation parameter $\rho = 0.5$. This change ensures that the synthetic data better reflects real-world scenarios where covariates are often correlated.
>
> Second, we extended the experiments to include more sophisticated models, such as kernelized robust regression (Section 3.2.3 along with Appendix A.1) and neural networks (Section 3.2.3 \& 5 along with Appendix A.2). Specifically, we used radial basis function (RBF) and polynomial kernels to capture nonlinear patterns in the data, as well as neural networks to evaluate our method's applicability to deep learning models.
>
> The results, presented in Figures 3, 5, and 6, demonstrate that LOO-StabCP performs robustly across all these settings. It maintained valid coverage while producing tight prediction intervals, even under complex data structures and model configurations. These findings highlight the versatility and practicality of our method beyond the simpler cases considered in the original submission. We appreciate your suggestion, as it has allowed us to showcase the broader applicability of LOO-StabCP and enhance the rigor of our empirical evaluations.

---

> ### Author Response · Authors · 2024-11-23
>
> ## Your Questions:
>
> ---
>
> 1. **Question:**
>    "*I think it is necessary to discuss more and carefully about applying LOO-StabCP to deep learning algorithms for reasons outlined in the weaknesses box.*"
>
> **Response:**
> We have addressed these points in detail in our earlier responses. Kindly refer to those responses, as we believe they address your questions. Thank you again for your thoughtful comments about this issue.
>
> ---
>
> 2. **Question:**
>    *As introduced in section 1, derandomization methods can address the issue of decreasing accuracy caused by randomly split but increase computational cost. I think it is better to demonstrate the performance of some derandomization method and compare it with LOO-StabCP to illustrate the differences between the two in all aspects.*
>
> **Response:**
> Thank you for pointing out this important aspect, which we had previously overlooked. In the revised manuscript, we explicitly addressed this in the very last part of Section 5 and provided a detailed discussion in Appendix B. In particular, we included a detailed comparison with two derandomization methods: which we named MM-SplitCP (Solari and Djordjilović, 2022) and EM-SplitCP (Gasparin and Ramdas, 2024).
>
> Our numerical experiments, presented in Figures 7 and 8, show that while derandomization methods effectively stabilize prediction intervals, they often produce conservative results with wider intervals. In contrast, LOO-StabCP achieves comparable stability without relying on random splits, avoiding additional randomness altogether. Moreover, LOO-StabCP produces tighter prediction intervals with valid coverage, maintaining competitive accuracy. Importantly, LOO-StabCP is computationally far more efficient, as it requires only a single model fit, whereas derandomization methods involve multiple fits across different splits, significantly increasing their computational cost.
>
> This comparison highlights the practical strengths of LOO-StabCP, particularly in scenarios where computational efficiency is critical. Your suggestion allowed us to address this key point, and we are grateful for your valuable feedback.
>
> ---
>
> 3. **Question:**
>    "*The authors compute stability-adjusted p-values for multiple selection in section 6. I think it is better to verify the validity of p-values in (7) and state it as a proposition for completeness.*"
>
> **Response:**
> This is a good question!
>
> We provide a short proof that the $p_j^{\rm LOO}$ defined in Equation (7) is indeed a valid p-value. Let $\tilde{f}$ denote an oracle $f$ trained on the data $X_1, \ldots, X_n$ and $X_{n+j}$. Then we know that the oracle p-value is defined as:
>
> $$
> p_j^{\rm oracle} = \frac{\sum_{i=1}^n \mathbf{1}\\{S(Y_i, \tilde{f}(X_i)) < S(Y_{n+j}, \tilde{f}(X_{n+j}))\\} + 1}{n+1},
> $$
>
> and is a valid p-value because the rank of $S(Y_{n+j}, \tilde{f}(X_{n+j}))$ among all oracle non-conformity scores is discrete uniform. To compare $p_j^{\rm oracle}$ with $p_j^{\rm LOO}$, we rely on the definition of LOO stability. Specifically, for all $i = 1, \ldots, n$ and $i = n+j$, we have:
> $$
> |S(Y_i, \tilde{f}(X_i)) - S(Y_i, \hat{f}(X_i))| \leq \tau_{i,j}^{\rm LOO},
> $$
> where $\hat{f}$ is the model trained without the $j$th data point. Under the null hypothesis, where $Y_{n+j} \leq c_j$, we also have:
> $$
> S(Y_{n+j}, \hat{f}(X_{n+j})) \leq S(c_j, \hat{f}(X_{n+j})).
> $$
> Combining these two inequalities, we get:
> $$
> S(Y_i, \tilde{f}(X_i)) < S(Y_{n+j}, \tilde{f}(X_{n+j}))
> \Rightarrow S(Y_i, \hat{f}(X_i)) - \tau_{i,j}^{\rm LOO} < S(c_j, \hat{f}(X_{n+j})) + \tau_{n+j,j}^{\rm LOO}.
> $$
> Thus, the indicator function satisfies:
> $$
> \mathbf{1}\\{S(Y_i, \tilde{f}(X_i)) < S(Y_{n+j}, \tilde{f}(X_{n+j}))\\} \leq \mathbf{1}\\{S(Y_i, \hat{f}(X_i)) - \tau_{i,j}^{\rm LOO} < S(c_j, \hat{f}(X_{n+j})) + \tau_{n+j,j}^{\rm LOO}\\}.
> $$
> Summing these indicator values, it follows that:
> $$
> p_j^{\rm oracle} \leq p_j^{\rm LOO}.
> $$
> Because $p_j^{\rm oracle}$ is a valid p-value under the null hypothesis, $p_j^{\rm LOO}$ also satisfies the validity property.
>
> As a side note, while addressing this question, we identified typos in Equations (6) and (7), which we have now corrected. Although these were typographical errors in the equations, they arose from shifting perspectives in representing the methodology when editing the draft for the first submission. Importantly, this did not affect our numerical results, as the implementation was consistent with the corrected representation. We sincerely thank you for your comment, which gave us the opportunity to identify and correct these typos.
>
> ---
>
> ## References
>
> [1] Aldo Solari and Vera Djordjilović. Multi split conformal prediction. *Statistics & Probability Letters*, 184:109395, 2022.
>
> [2] Matteo Gasparin and Aaditya Ramdas. Merging uncertainty sets via majority vote. *arXiv preprint arXiv:2401.09379*, 2024.

---

> > ### Comment · Reviewer_Rbu1 · 2024-11-27
> >
> > Thank you for the response. I will keep my score and tend to accept.

---

> > > ### Author Response · Authors · 2024-11-27
> > >
> > > Thank you for your thoughtful review and for maintaining a positive opinion about our work. In particular, your suggestion to conduct additional numerical experiments greatly contributed to enhancing the persuasiveness of our paper. We plan to further develop our framework by conducting several follow-up studies, and your insight will be a great help in this regard.
> > >
> > > We sincerely appreciate it!

---

### Official Review · Reviewer_3RnT · 2024-11-06

**Soundness:** 3
**Presentation:** 3
**Contribution:** 3
**Rating:** 6
**Confidence:** 2

**Summary:**

This paper proposes Leave-One-Out Stable Conformal Prediction, a novel method to speed up full conformal using algorithmic stability without sample splitting.

**Strengths:**

This paper is clearly written and easy to follow.

**Weaknesses:**

- Typos and minor issues
	- L103: notation [1:m] is not defined
	- L190: the objective function (is) often highly nonconvex

**Questions:**

- L33: Do we require $\mathcal{Y} \subseteq \mathbb{R}$? Do we require the marginal distribution $P_Y$ is continuous for the non-conformity scores to be uniformly distributed (L81)?
- L34: Is D_test drawn from the same distribution $P_{X,Y}$ as D? Is it iid drawn? If yes, then is it equivalent to consider a single test example (e.g., Equation (1) only need to hold for one data point instead of for all $j \in [m]$)? If not, then why all non-conformity scores are exchangeable (L78)?
- L81: Does this hold for any alpha in [0,1]? How do we obtained this from the fact that the rank is uniformly distributed over $\{1, \dots, n+1\}$?
- L81:` $\mathcal{Q}_{1-\alpha}$` is not defined. Is it lower quantile function `$\mathcal{Q}_{p}:= \inf \{x: F(x) \geq p\}$`?
- L86 (Equation 2)" Shouldn't $1-\alpha$ be slightly increased to $\frac{\lceil (1-\alpha)(n+1) \rceil}{n}$ for this to hold in finite sample? See Equation (19) of https://www.stat.berkeley.edu/~ryantibs/statlearn-s23/lectures/conformal.pdf.

---

> ### Author Response · Authors · 2024-11-23
>
> We greatly appreciate your positive comments. Here is a point-by-point response to your concerns.
>
> ---
> ### Typos and minor issues:
>
> - **Comment:**
>   *L103: notation $[1:m]$ is not defined*
>
>   **Response:**
>   Thank you for this comment. We have changed the notation to a more common notion $[m]$ and defined it at the beginning of our recap of full conformal.
>
> - **Comment:**
>   *L190: the objective function (is) often highly nonconvex*
>
>   **Response:**
>   Thank you, done.
>
> ---
>
> ### Questions:
>
> - **Comment:**
>   *Do we require $\mathcal{Y} \subseteq \mathbb{R}$? Do we require the marginal distribution $P_Y$ is continuous for the non-conformity scores to be uniformly distributed (L81)?*
>
>   **Response:**
>   Thank you for this comment. Yes, we do require $\mathcal{Y} \subseteq \mathbb{R}$ as we do not consider complex-valued responses. We omitted this notion because it is a common assumption in conformal prediction literature.
>
>   Your question regarding the continuity of $P_Y$ is insightful and interesting. Inspecting our approach, we did not explicitly use the continuity assumption (in the development of our method, or in theoretical analysis). We noticed that we did state that we assume $P_{X,Y}$ is continuous. This is unnecessary and we have removed it from the revised version. That being said, it is true that some $P_{X,Y}$ and $f$ configuration may lead to large stability bounds, thus conservative prediction sets. Also, for simplicity, throughout the paper, we have been focusing on one specific choice of the non-conformity score, written in over-simplified notation, that $S = |y-f(x)|$. This might not be the choice of non-conformity scores for classification. But we feel that may go beyond the scope of this paper and did not discuss.
>
>   In the conformal prediction literature, there is a classical technique for breaking ties by adding a small amount of artificial random noise (independent of anything else) to each non-conformity score, see Romano et al. (2020) "Classification with Valid and Adaptive Coverage." Therefore, the (discrete) uniform distribution of the rank is not so much of a concern as it might seem. But it's a good question!
>
>
> - **Comment:**
>   *Is $\mathcal{D}\_{\text{test}}$ drawn from the same distribution $P_{X,Y}$ as $\mathcal{D}$? Is it iid drawn? If yes, then is it equivalent to consider a single test example (e.g., Equation (1) only need to hold for one data point instead of for all $j \in [m]$)? If not, then why all non-conformity scores are exchangeable?*
>
>   **Response:**
>   Thank you for pointing this out. We have added a clarification that the test data (including the unobserved responses) are also i.i.d. from the same distribution $P_{X,Y}$.
>
> - **Comment:**
>   *Does this hold for any $\alpha$ in $[0,1]$? How do we obtain this from the fact that the rank is uniformly distributed over $\\{1,\dots,n+1\\}$?*
>
>   **Response:**
>   Yes, the result holds for any $\alpha \in [0,1]$. To elaborate, the key is that the rank of the test score among the $n+1$ non-conformity scores is uniformly distributed over the discrete set $\\{1, \dots, n+1\\}$. This means that the rank has an equal probability of occupying any of these $n+1$ positions, with probability $\frac{1}{n+1}$ for each. Now, for the test score to fall within the $(1-\alpha)$-quantile, its rank must be less than or equal to $(1-\alpha)(n+1)$. Since the rank is discrete, this corresponds to the smallest integer greater than or equal to $(1-\alpha)(n+1)$, which is $\lceil(1-\alpha)(n+1)\rceil$. Summing the probabilities up to this rank gives the coverage probability, $\frac{\lceil(1-\alpha)(n+1)\rceil}{n+1} \geq 1-\alpha$.
>
>
> - **Comment:**
>   *$\mathcal{Q}\_{1-\alpha}$ is not defined. Is it lower quantile function $\mathcal{Q}_{p}:= \mathrm{inf} \\{x: F(x) \geq p\\}$?*
>
>   **Response:**
>   Thank you for pointing this out. We have added the definition of $\mathcal{Q}$ immediately following its first appearance.
>
>
> - **Comment:**
>   *In (2), shouldn't $1-\alpha$ be slightly increased to $(1-\alpha)$-quantile is $\frac{\lceil(1-\alpha)(n+1)\rceil}{n}$ for this to hold in finite sample?*
>
>   **Response:**
>   Thank you for the insightful comment. This is a bit subtle—let us explain. There is a difference between Equation (19) in the material you shared and Equation (2) in our manuscript. Equation (19) computes the quantile based only on $n$ data points, excluding the test example, while our Equation (2) includes $\infty$. In the former, a correction is needed because one of the original $n+1$ data points is excluded, while in our formulation, the inclusion of $\infty$ allows us to compute the quantile directly over $n+1$ points, eliminating the need for such a correction.
>
> ---
>
> ### References:
>
> [1] Yaniv Romano, Evan Patterson, and Emmanuel J. Candès. Classification with valid and adaptive coverage. *Advances in Neural Information Processing Systems*, 33:3581–3591, 2020.

---

> > ### Comment · Reviewer_3RnT · 2024-11-28
> >
> > I confirm that I have read the response, which answered my questions. I will keep my score and tend to accept.

---

> > > ### Author Response · Authors · 2024-11-28
> > >
> > > Thank you for keeping your positive view of our work. We look forward to building on this framework through further studies, and your input will be invaluable as we move forward.
> > >
> > > We truly appreciate your support!

---

### Author Response · Authors · 2024-11-23
**Revised Manuscript and Responses**

Dear AC and reviewers,

Thank you so much for your invaluable comments on our work!
We have updated the paper with new theoretical and numerical results, along with point-by-point responses to your concerns.
We look forward to the discussion.

Thank you! \
Anonymous Authors

---

### Author Response · Authors · 2024-11-23
**Reply to Common Concern: More Applications/Examples**

We greatly appreciate the constructive comments from the reviewers!
We recognize that there exists a common concern shared by all of you. We will respond to this question here to avoid repetitive individual replies on the same issue.

---

### The shared comment is:

- **What other applications/examples does our method apply to?**

Specifically, we understand that reviewers have the following related/elaborating comments:

1. Reviewers Rbu1, LXUa, and VLB9 all urged us to explore more applications of our results, specifically emphasizing the application to neural networks.
2. Reviewer VLB9 felt that RLM and SGD in our paper are "very limited" applications.
3. Reviewer Rbu1 mentioned that conditions such as Lipschitzness and convexity required by Theorem 2 (RLM) and Theorem 3 (SGD) may be difficult to verify, and the LOO stability bound may be difficult to calculate in practice.

---

### Added Applications/Examples
In response to your request, we have added the following applications/examples to the revised version of this paper:

- Kernel method (closely related to RLM)
- Neural networks (via our new theorem for SGD that does not require convexity)
- Bagging, which includes random forest as a special case

---

### Addressing Concerns

**For 1.**
In the rebuttal revision, we present a new theoretical result, applying our method to SGD with potentially non-convex objective functions, see Theorem 4 (highlighted in blue).

**For 2.**
Our understanding is that RLM and/or SGD can serve as the basic tools in many learning problems. For example, RLM covers many likelihood-based methods in statistical learning. In the revised version, we also raised the well-known kernel method as a special case of RLM, demonstrating its generality. Also, neural networks with $\ell_2$ regularization can also be formulated in the format of RLM. As for SGD, the application is even wider since it is an optimization tool that can serve any learning task that involves optimization. For example, it has been a popular optimization tool for fitting deep neural networks (DNN). In our new theory, we apply SGD to analyze neural networks (Theorem 4).

In the revised version, our analysis for the newly added application bagging is different than RLM and SGD, see Theorem 5.

**For 3.**
In this paper, we adopt the deterministic definition of algorithmic stability as in Ndiaye (2022), which considers the worst case. Our approach is thus different from the distributional notion of algorithmic stability as in Barber et al. (2021), which may require some knowledge (e.g., light-tail) about the data distribution. Consequently, these Lipschitz, smoothness and convexity constants, whichever applicable, are the properties of the deterministic objective functions.

For example, to obtain those constants in the case of robust linear regression that we used, we could directly compute derivatives and find the bounds. Similarly, for neural networks composed of simple functions, the bounds can still be derived using the chain rule. In these examples, all needed components of the stability bound can be directly derived or read from the problem set-up and the method's formulation.

Meanwhile, we do agree with you that there exists no universal analytical formula for stability bounds for any prediction method $f$. This is the reality not only for our method, but for all algorithmic stability results that we know of  (Soloff et al, 2024; Liang et al, 2023; Ndiaye, 2022; Wang et al, 2023). The stability bound formula for each application/example needs to be developed by researchers. We deem the development of these bounds for more prediction methods as an intriguing venue for future work.

---

> ### Author Response · Authors · 2024-11-23
> **(Continued)**
>
> ### Remarks on Neural Networks
>
> Furthermore, we would like to add a few important remarks regarding the application of our method to DNN.
>
> The main challenge in applying our method to DNN's is the convexity constraint in our Theorems 2 \& 3.
> DNN's are typically highly non-convex.
> Our idea to address DNN is not to analyze the shape of the objective function at the eventually convergent parameter values, which is super-complicated.
> Instead, we analyze the algorithm SGD as a popular optimizer for deep neural networks.
> This led to our new theoretical result, i.e., Theorem 4, which is proved in a different way than its convex counterpart (Theorem 3).
>
> Nonetheless, the extension of SGD to non-convex functions is not a free lunch.
> The price is that the mathematically rigorous stability bound in Theorem 4 may be quite conservative.
>
> Recall that we have derived a stability bound in the original submission for SGD applicable to convex functions, which has a much tighter stability bound.
> We numerically tested its performance on neural networks, and it seemed to perform well.
> See Figure 5.
> It achieves desired coverage rate while showing competitive accuracy.
> Computation-wise, it is much faster than RO-StabCP.
>
> Therefore, a very intriguing future work is to derive tighter stability bounds for non-convex SGD.
> The task seems challenging and may require significant additional endeavors.

---

> > ### Author Response · Authors · 2024-11-23
> > **References**
> >
> > [1] Rina Foygel Barber, Emmanuel J. Candès, Aaditya Ramdas, and Ryan J. Tibshirani. Predictive inference with the jackknife+. *The Annals of Statistics*, 49(1):486–507, 2021. doi: 10.1214/20-AOS1965. https://doi.org/10.1214/20-AOS1965.
> >
> > [2] Eugene Ndiaye. Stable conformal prediction sets. In *International Conference on Machine Learning*, pp. 16462–16479. PMLR, 2022.
> >
> > [3] Jake A Soloff, Rina Foygel Barber, and Rebecca Willett. Bagging provides assumption-free stability. *Journal of Machine Learning Research*, 25(131):1–35, 2024.
> >
> > [4] Ruiting Liang and Rina Foygel Barber. Algorithmic stability implies training-conditional coverage for distribution-free prediction methods. *arXiv preprint arXiv:2311.04295*, 2023.
> >
> > [5] Yan Wang, Huaiqing Wu, and Dan Nettleton. Stability of random forests and coverage of random-forest prediction intervals. *Advances in Neural Information Processing Systems*, 36:31558–31569, 2023.

---

### Meta-Review · Area_Chair_CBnE · 2024-12-24

**Metareview:**

Computing full conformal prediction can be challenging if applied to new test points because all the evaluations need to be repeated for each new test points. This paper leverage algorithmic stability bound that are independent to the left out points and derive faster algorithms.

One weakness I might foresee is that while improving computational time, this independence to the test point will back fire on the adaptability of the cp sets. For example, in heteroscedastic setting, one want to depends on the points. It should be nice if authors could add more experiments on these lines.

**Additional Comments On Reviewer Discussion:**

Reviewers generally acknowledge its practical contribution and solid theoretical foundations, emphasizing its advantages in handling multiple predictions efficiently while maintaining valid coverage. However, they raise concerns about its novelty and limited application examples. Specifically, while LOO-StabCP demonstrates significant computational savings compared to RO-StabCP, some reviewers feel the extension from replace-one to leave-one-out stability is straightforward and lacks substantial innovation.

The authors responded by emphasizing the conceptual leap of leave-one-out stability as a non-obvious methodological innovation, challenging conventional conformal prediction doctrines. They expanded the paper with new applications to neural networks, kernel methods, and bagging, and demonstrated the method's versatility across broader scenarios, including non-convex settings. While the additional results and applications convinced some reviewers to accept or maintain positive ratings, one reviewer remained skeptical about the novelty but acknowledged the value of the added contributions. The overall sentiment reflects recognition of LOO-StabCP's practical relevance and computational efficiency, balanced against debates about its originality.


- Reviewer 3RnT: Positive overall, supports acceptance with minor suggestions for improvement.
- Reviewer Rbu1: Marginally positive, leans toward acceptance after the authors addressed concerns with expanded experiments and additional analyses.
- Reviewer LXUa: Strongly positive, advocates for acceptance following the revisions and enhanced experimental evaluations.
- Reviewer VLB9: Borderline, appreciates the revisions and additional results but maintains concerns regarding the novelty of the contribution.

---

### Decision · Program_Chairs · 2025-01-22

Accept (Poster)